# Understanding End-to-End Model-Based Reinforcement Learning Methods as Implicit Parameterization

**Clement Gehring**
Electrical Engineering and Computer Sciences
Massachusetts Institute of Technology
clement@gehring.io

**Kenji Kawaguchi**
Center of Mathematical Sciences and Applications
Harvard University
kkawaguchi@fas.harvard.edu

**Jiaoyang Huang**
Courant Institute of Mathematical Sciences
New York University
jh4427@nyu.edu

**Leslie Pack Kaelbling**
Electrical Engineering and Computer Sciences
Massachusetts Institute of Technology
lpk@csail.mit.edu

## Abstract

Estimating the per-state expected cumulative rewards is a critical aspect of reinforcement learning approaches, however the experience is obtained, but standard deep neural-network function-approximation methods are often inefficient in this setting. An alternative approach, exemplified by value iteration networks, is to learn transition and reward models of a latent Markov decision process whose value predictions fit the data. This approach has been shown empirically to converge faster to a more robust solution in many cases, but there has been little theoretical study of this phenomenon. In this paper, we explore such implicit representations of value functions via theory and focused experimentation. We prove that, for a linear parametrization, gradient descent converges to global optima despite non-linearity and non-convexity introduced by the implicit representation. Furthermore, we derive convergence rates for both cases which allow us to identify conditions under which stochastic gradient descent (SGD) with this implicit representation converges substantially faster than its explicit counterpart. Finally, we provide empirical results in some simple domains that illustrate the theoretical findings.

## 1 Introduction

Whether done by averaging over trajectories or some intricate fitting procedure, estimating the per-state expected cumulative rewards is a critical aspect of every reinforcement learning (RL) approach. Efficiently learning these values in large complex domains still remains a challenge despite advances in deep learning. One contributing factor comes from the difficulty of scaling model-based methods which model the transition probabilities and rewards directly. While known to be sample efficient, these methods have failed to fully leverage recent advances in deep learning, forcing the use of less efficient but more scalable model-free methods which try to learn the values directly. Even when achieving low one-step prediction error, models learned with large highly expressive

35th Conference on Neural Information Processing Systems (NeurIPS 2021).

estimators struggle to produce good policies or accurate value estimates in part due to a mismatch in the objective [11] or compounding errors [24, 22]. To address this, several recent methods propose to directly optimize predictions derived from the model and leverage modern automatic differentiation frameworks to propagate errors back to the model parameters [23, 18, 15, 19, 6]. However, despite promising results shown by these methods, often referred to as *end-to-end* model-based methods, there is little formal justification for this idea beyond intuitive arguments.

In many cases, this approach is motivated by the goal of varying the model parameters based on the current state or observation. Empirical results suggest that such end-to-end model-based methods are capable of better generalization when observations can be used to formulate smaller subproblems, for example, by providing a map of the world [23], a description of a goal [19], or the high-level action being followed [15]. However, as we will show later, even without input dependent models, this indirect, or *implicit*, representation of values drastically alters the dynamics of gradient descent and, thus, the inductive bias, in interesting ways.

To better understand how these end-to-end model-based methods compare to more direct methods, we formulate these ideas as different parameterization of the same function class. Specifically, by formulating an end-to-end model-based approach as an implicit parameterization of a linear function, we can directly quantify differences between this implicit approach to the more common trivial explicit parameterization where linear weights are stored and learned directly.

Concurrent to our work was a recent paper [14] which focuses on implicit differentiation applied to end-to-end model based methods. This work compares this approach to model-based methods which maximize likelihood. Although we focus on the comparison between end-to-end methods and other direct methods, we view this related work as highly relevant and complementary with ours. Both our work and theirs aim to better understand the properties of end-to-end model-based methods.

Our contributions are primarily theoretical and aim to provide a theoretical account of the performance of end-to-end model-based methods. To help in this matter, we also provide some empirical results in simple illustrative problems which serve to demonstrate properties derived from our analysis. We start by showing that although the implicit formulation defines a non-linear and non-convex optimization problem, the implicit linear weights still converge to a global optimum. We continue by showing that stochastic gradient descent with the implicit parameterization has appealing properties when the residuals exhibit certain properties. Specifically, we derive the expected changes in the loss and show a surprisingly fast convergence rate along a specific component of the residual. We then use these results to discuss conditions under which end-to-end methods might be preferable. We conclude our theoretical contributions by quantifying the variance-mean ratio updates under both parameterization and establishing conditions under which the implicit parameterization exhibits significantly better properties than its explicit counterpart. Finally, we conclude with some empirical results.

## 2   Framework

While our theoretical results consider a supervised learning setting, the motivation for this work comes from the policy evaluation problem. In this setting, we assume there is an MDP, $\mathcal{M} = \{\mathcal{S}, \mathcal{A}, r_a, T_a\}$, with a state space $\mathcal{S}$ whose dynamics for each action $a \in \mathcal{A}$ are described by the transition probabilities $T_a$ generating rewards as described by $r_a$. We are interested in learning the discounted value function for a given policy $\pi$ which is defined as the per-state expected cumulative discounted rewards. Formally, given a discount factor $\gamma \in [0, 1)$ and state $s \in \mathcal{S}$,

$$V^\pi(s) = \mathbb{E}_{A_t \sim \pi(S_t)} \left[ \sum_{t=0}^{\infty} \gamma^t R_{(t)} \,\Bigg|\, S_0 = s \right] = \mathbb{E}_{A_0 \sim \pi(s)} \left[ R_0 + \gamma V^\pi(S_1) \mid S_0 = s \right], \quad (1)$$

where the second equality leverages the Markov assumption to define the values recursively.

### 2.1   Linear function approximation

We investigate the differences between parameterizations in the context of linear functions of the form $\hat{V}(s) = \phi(s)^\top \theta$, where $\phi(s) \in \mathbb{R}^k$ is some fixed, possibly non-linear, encoding of the state $s \in \mathcal{S}$ and $\theta \in \mathbb{R}^k$ represents the linear coefficients. This type of function approximator is common in reinforcement learning and can be seen as a generalization of the discrete tabular setting, i.e., $\phi(s)$

corresponds to a one-hot encoding, and has yielded insight in the past, for example, by highlighting temporal difference learning convergence issues [1] or by relating Bellman error to model error [16]. Additionally, in settings where domain knowledge can be used to generate useful features, these linear approximations can be significantly more efficient to learn and evaluate, making them not only theoretically interesting, but often practical as well.

To allow us to compare different parameterizations, we introduce a reparameterization function $g$ which maps parameters $\psi \in \Psi$ to linear feature weights $\theta \in \Theta$, i.e., $\theta := g(\psi)$ and define linear estimators of the form:

$$\hat{V}_g(s; \psi) = \phi(s)^\top g(\psi) = \phi(s)^\top \theta.$$

Note that while $\hat{V}_g$ is always linear in $\phi(s)$ and $\theta$, it is not necessarily linear with respect to the parameters $\psi$. For this reason, the dynamics of gradient descent can differ significantly across different parameterization despite all defining equally expressive function classes (if $g$ spans $\mathbf{R}^n$).

## 2.2 Implicit parameterization

In order to explore the properties of end-to-end model-based methods, we consider a simple instance of such methods whose model corresponds to a Markov reward chain, i.e., an MDP with a singleton action. We consider the case where the rewards are encoded as a vector $w$ and the transition probabilities are kept as a matrix of unnormalized log-probabilities, $F$, such that $\sigma(F)$ corresponds to a row-stochastic matrix, i.e., $\sigma$ applies the softmax function on each row. With the probabilities encoded in this way, we can treat $F$ as an unconstrained matrix which simplifies both the optimization problem and our analysis. Finally, we define the linear weights $\hat{\theta} := g((w, F))$ to be the state values of the Markov reward chain defined by $(w, F)$.

We note that this formulation closely resembles that of value iteration networks (VIN) [23] with a few differences. Although VIN allows the model parameters to be conditioned on the state, we wish to better understand the effect of representing values implicitly, as model parameters, without any conflating effect introduced by using state-dependent models. Note that any insight in this simpler framework is likely relevant to the state-dependent model setting since the gradients differ only by the Jacobian relating model parameters' differentials to learned parameters' differentials, i.e., the chain rule. Additionally, while VIN consider an internal action space similar to the action space of the true underlying MDP, we limit ourselves to the a singleton action in order to keep the analysis tractable. We provide a brief explanation for why insight in the singleton case is relevant to the general case and provide some supporting empirical results in the appendix.

We assume the linear feature weights, $\hat{\theta}$, and the model parameters, $\psi$, satisfy some constraint $h(\hat{\theta}; \psi) = 0$ which can be used to implicitly define the function, $g$, which maps model parameters to feature weights, i.e., $g(\psi) = \hat{\theta}$. This implicit formulation allows us to analyse how $\psi$ and $\hat{\theta}$ relate without needing to consider the algorithm used to compute $\hat{\theta}$. For the Markov reward chain case previously described, we can use (1) to define the constraints as

$$h(\hat{\theta}; w, F) = w + \eta\sigma(F)\hat{\theta} - \hat{\theta}, \tag{2}$$

where $\eta \in [0, 1)$ is the *internal* discount factor for the modelled Markov reward chain. Note, that this discount factor $\eta$ is different than the discount factor $\gamma$ which defines the true discounted values. We intentionally use a different symbol to avoid any confusion later on. Also, for clarity, we allow ourselves to abuse the notation slightly by unpacking the parameters $\psi$ into a vector, $w$, and matrix, $F$, analogous to the rewards and the unnormalized transition log-probabilities, respectively.

## 2.3 Implicit differentiation

Under appropriate assumptions of continuity, an implicit parameterization can be differentiated just like any other explicit parameterization through the use of the implicit function theorem. Formally, for an implicit parameterization $g$, the derivatives can be defined solely through the derivatives of the constraint, $h(\theta; \psi) = 0$. That is, the constraint of $0 = h(\theta; \psi) = h(g(\psi); \psi)$ (where $\theta = g(\psi)$) implies that $0 = \frac{\partial h}{\partial \theta} \frac{\partial g}{\partial \psi} + \frac{\partial h}{\partial \psi}$ (by the chain rule), which can be rearranged into

$$\frac{\partial g}{\partial \psi} = -\left(\frac{\partial h}{\partial \theta}\right)^{-1} \frac{\partial h}{\partial \psi}.$$

Note that this results in differentiation rules that are independent of how the constraints were solved. In contrast, many methods like VIN will differentiate through the solver's computation, e.g., several iterations of value iteration. This approach is often only asymptotically equivalent to the implicit differentiation formulation and will otherwise alter the gradient dynamics.

## 3 Dynamics of gradient descent

We start by exploring the dynamics of gradient descent under the implicit parameterization induced by (2). We show that the gradient descent updates under the implicit parameterization are equivalent to preconditioned updates under the explicit parameterization. Furthermore, we show that despite no longer being a convex optimization problem, the gradient dynamics of the implicit parameterized objective function converges to a global optimum.

To keep the notation clear, we reserve $\theta$ to represent parameters of the trivial explicit parameterization in which the reparameterization function is the identity, $g_{explicit}(x) = x$, and $\hat{\theta}$ to represent the linear weights under an implicit parameterization, e.g., $g_{implicit}(w, F) = \hat{\theta}$ where $\hat{\theta}$ is the solution to the constraint (2). For the remainder of this paper, we will refer to these two parameterizations as the explicit and implicit parameterization, respectively.

We assume we are given a set of $n$ datapoints $\mathcal{D} = \{(x_i, y_i) : i \in \{1, \ldots, n\}\}$ and a loss $L$ defined on the linear weights such that

$$L(\theta) = \frac{1}{2} \sum_i^n (\phi(x_i)^\top \theta - y_i)^2 = \frac{1}{2} \sum_i^n r_i(\theta)^2, \tag{3}$$

where $r_i(\theta)$ is the residual of the predicted value of $x_i$ using the linear weights $\theta$ and $\phi$ is a fixed function mapping data points, $x_i$, to feature vectors. Additionally, we assume a fixed sequence of learning rates $(\alpha_{(t)})_{t=0}^\infty$ is used. Let the sequence $(\theta_{(t)})_{t=0}^\infty$ represent the linear weights obtained by using gradient descent to optimize $L$ under the explicit parameterization with initial weights $\theta_{(0)}$ and, similarly, let $(\hat{\theta}_{(t)})_{t=0}^\infty$ be the sequence of implicitly parameterized linear weights obtained by optimizing $L(g_{implicit}(w_{(t)}, F_{(t)}))$ with corresponding parameters $(w_{(t)})_{t=0}^\infty$ and $(F_{(t)})_{t=0}^\infty$, and initial parameters $(w_{(0)}, F_{(0)})$. Following previous works on gradient dynamics of deep learning models [17, 4, 5, 8], we also consider the corresponding gradient dynamics $\frac{d}{dt}\theta_{(t)}$ and $\frac{d}{dt}\hat{\theta}_{(t)}$ under the gradient flow (where $\frac{d}{dt}\hat{\theta}_{(t)}$ is induced by the dynamics of $w_{(t)}$ and $F_{(t)}$).

We start our analysis by presenting our results describing the gradient dynamics for both parameterizations. This will allow us to compare the two and start building a better understanding of the effects the implicit parameterization has on the gradient dynamics. First, we define $A_{(t)} = I - \eta\sigma(F_{(t)})$ and note that because $\sigma(F_{(t)})$ always forms a stochastic matrix, we can explicitly write out the implicitly parameterized linear weight, $\hat{\theta}_{(t)} = A_{(t)}^{-1} w_{(t)}$, enabling us to show the following theorem.

**Theorem 1.** Let $P_{(t)} = A_{(t)}^{-1} \left( \eta^2 D_{(t)} + I \right) (A_{(t)}^{-1})^\top$ where $D_{(t)} = \mathrm{diag}(\mathbf{d}_{(t)})$ and $[\mathbf{d}_{(t)}]_k = \sum_i \left( \sum_j \frac{\partial\sigma(F_{(t)})_{ki}}{\partial[F_{(t)}]_{kj}} \theta_j \right)^2$, then $\lambda_{\min}(P_{(t)}) \geq c$ for some time-independent constant $c > 0$, and the following holds:

$$\frac{d}{dt}\theta_{(t)} = -\nabla L(\theta_{(t)}) \qquad and \qquad \frac{d}{dt}\hat{\theta}_{(t)} = -P_{(t)}\nabla L(\hat{\theta}_{(t)}). \tag{4}$$

We provide the proofs for all our results in the supplementary material. From this result, we see that the updates to the linear weights under the implicit parameterization resemble that of preconditioned gradient descent under the explicit parameterization where $P_{(t)}$ acts as a preconditioner which transforms the gradient before updates. Ideally, the preconditioner would be chosen such that it accelerates convergence for some class of problems, for example, by efficiently approximating the inverse Hessian thus resulting in updates approximating Netwon's method. This potential to greatly accelerate convergence under the right circumstances is why our analysis will revolve around understanding the properties of $P_{(t)}$.

We begin by noting that $P_{(t)}$'s positive definiteness implies that the weights of the implicit parameterization converge to a global minimum from any initialization.

**Corollary 1.** *For any initialization $\hat{\theta}_{(0)}$, the gradient dynamics $\frac{d}{dt}\hat{\theta}_{(t)}$ converges to a global minimum.*

It is important to note that the loss landscape under the implicit parameterization is generally non-linear and non-convex. As a consequence, convergence results applicable to linear regression cannot be trivially applied to our setting; additional properties of the optimization dynamics are required, such as those presented in Theorem 1. Thus, we proved a non-trivial global convergence in non-convex optimization for reinforcement learning, whereas related works studied it for deep learning [7, 9, 10, 26].

### 3.1 Properties of stochastic gradient descent

We now turn our focus towards stochastic gradient descent (SGD), e.g., gradient descent on sampled mini-batches of training data, and present some additional properties of both parameterizations considered. In this section, we first define some additional notation and update rules to accommodate the SGD setting. We then quantify the expected improvement of the loss and show that both cases have comparable convergence rates.

We start by defining a sequence of random averaged loss functions $L_{(0)}, L_{(1)}, \ldots$, corresponding to the loss $L$ defined in (3) evaluated on a random subset of the data $\mathcal{D}$. Formally,

$$L_{(t)}(\theta) = \frac{1}{2|\mathcal{D}_{(t)}|} \sum_{(x,y)\in\mathcal{D}_{(t)}} (\phi(x)^\top\theta - y)^2,$$

where $\mathcal{D}_{(t)}$ is a random subset of the data $\mathcal{D}$ of size $k$ drawn without replacement. In this setting, gradients are taken just as in gradient descent but using $L_{(t)}$ instead of the full loss.

Before presenting our results quantifying the dynamics of SGD with the explicit and implicit parameterization, we need the following assumptions on the feature matrix.

**Assumption 1.** *Using the notation $X \lesssim Y$ to mean $X < cY$ for some constant $c > 0$, we assume the features $\phi(x_i)$ satisfy*

- $\|\phi(x_i)\|_2 \lesssim 1$ *for $1 \le i \le n$.*

- *Let $X = [\phi(x_1), \phi(x_2), \cdots, \phi(x_n)]$, the sample covariance matrix $X^\top X$ has bounded norm $\|X^\top X\|_2 = \mathcal{O}(1)$, and $\|X^\top \mathbf{1}\|_2^2 \gtrsim n$.*

**Theorem 2.** *Given a sequence of learning rates $(\alpha_{(t)})_t$, for SGD with the explicit parameterization, the expected decrease of the loss function satisfies*

$$\mathbb{E}[(L(\theta_{t+1}) - L(\theta_{(t)}))|\theta_{(t)}] = -\frac{\alpha_{(t)}}{n}\left\|\mathbf{r}(\theta_{(t)})\right\|_2^2 + \mathcal{O}(\alpha_{(t)}^2) \gtrsim -\frac{\alpha_{(t)}}{n}L(\theta_{(t)}) + \mathcal{O}\left(\alpha_{(t)}^2\right),$$

*where $\mathbf{r}(\theta) = \sum_i r_i(\theta)\phi(x_i)$.*

If $\alpha_t = \mathcal{O}(1)$, in expectation, the convergence rate for the SGD with explicit parametrization is $\mathcal{O}(1/n)$ which makes it slow, taking $\mathcal{O}(n)$ steps to converge. Note that this result isn't novel and follows from standard results of convex optimization [3], but is stated here to facilitate comparisons. For the implicit parameterization, we recall from Theorem 1 (and Lemma **??**) that the GD dynamics are given explicitly in terms of the matrix $P_{(t)}$. We state formally a similar result for the expected SGD dynamics:

**Theorem 3.** *Given a sequence of learning rate $(\alpha_{(t)})_t$, for SGD with the implicit parameterization, the expected decrease of the loss function satisfies*

$$\mathbb{E}[(L(\hat{\theta}_{t+1}) - L(\hat{\theta}_{(t)}))|\hat{\theta}_{(t)}] = -\frac{\alpha_{(t)}}{n}\left\langle P_{(t)}\mathbf{r}(\hat{\theta}_{(t)}), \mathbf{r}(\hat{\theta}_{(t)})\right\rangle + \mathcal{O}(\alpha_{(t)}^2),$$

*where $\mathbf{r}(\theta) = \sum_i r_i(\theta)\phi(x_i)$.*

### 3.2 Conditions favoring implicit parameterization

An important takeaway of our results so far is that the matrix $P_{(t)}$ is critical in understanding the effects of using such an implicit parameterization, and, more importantly, in understanding when it would be expected to outperform its explicit counterpart. For this reason, we first examine some

properties of $P_{(t)}$. To this end, we first must highlight some general properties of row-stochastic matrices.

Recall that, by construction, $\sigma(F_{(t)})$ is an irreducible row-stochastic matrix for any finite $F_{(t)}$. As a result, the Perron-Frobenius theorem tells us that the largest eigenvalue (in absolute value) of $\sigma(F_{(t)})$ is 1 with all other eigenvalues strictly smaller and there exists an eigenvector $\mathbf{v}_{(t)}$ such that

$$\sigma(F_{(t)}) = \mathbf{1}\mathbf{v}_{(t)}^\top + M_{(t)}, \quad \mathbf{v}_{(t)}^\top \mathbf{1} = 1, M_{(t)}\mathbf{1} = 0, \quad \mathbf{v}_{(t)}^\top M_{(t)} = 0,$$

where $\mathbf{v}_{(t)}$ can be interpreted as the stationary distribution of a Markov chain with transition probabilities $\sigma(F_{(t)})$. This observation allows us to decompose $A_{(t)}^{-1}$ and identify its dominant terms:

$$A_{(t)}^{-1} = \left(I - \eta\sigma(F_{(t)})\right)^{-1} = I + \sum_{k \geq 1} \eta^k \mathbf{1}\mathbf{v}_{(t)}^\top + \sum_{k \geq 1} \eta^k M_{(t)}^k = \frac{\eta}{1-\eta}\mathbf{1}\mathbf{v}_{(t)}^\top + \sum_{k \geq 0} \eta^k M_{(t)}^k.$$

Since all eigenvalues of $M_{(t)}$ are strictly less than 1, we see that for an arbitrary vector $\mathbf{x}$

$$A_{(t)}^{-1}\mathbf{x} = \frac{1}{1-\eta}\mathbf{1}\left(\mathbf{v}_{(t)}^\top \mathbf{x}\right) + \mathcal{O}(1), \tag{5}$$

as $\eta$ approaches 1. This decomposition of $A_{(t)}^{-1}$ allows us to see that the vectors $\mathbf{1}$ and $\mathbf{v}_{(t)}$ also dominate the dynamics induced by $P_{(t)}$. Specifically, we see that

$$P_{(t)} = A_{(t)}^{-1}\left(\eta^2 D_{(t)} + I\right)(A_{(t)}^{-1})^\top = \frac{1}{(1-\eta)^2}\mathbf{1}\mathbf{v}_{(t)}^\top \left(\eta^2 D_{(t)} + I\right)\mathbf{v}_{(t)}\mathbf{1}^\top + \mathcal{O}\left(\frac{1}{1-\eta}\right).$$

Recall that the expected improvement under the implicit parameterization is dominated by $\langle P_{(t)}\mathbf{r}(\hat{\theta}_{(t)})\mathbf{r}(\hat{\theta}_{(t)})\rangle$. This means that our results suggest that the component of the residuals, $\mathbf{r}(\hat{\theta}_{(t)})$ along the $\mathbf{1}$ direction, i.e., $\mathbf{1}^\top \mathbf{r}(\hat{\theta}_{(t)})$, plays a critical role in determining the performance under the implicit parameterization. We use this observation to better understand under which conditions we might expect the implicit parameterizations to perform better. We start by formally quantify the dynamics of the component of the vector of residual along this direction with the following theorem.

**Theorem 4.** *Let $r^\|(\hat{\theta})$ be the component of the vector of residuals parallel to the $\mathbf{1}$ direction, i.e., $\mathbf{r}(\hat{\theta}) = r^\|(\hat{\theta})\mathbf{1} + \mathbf{r}^\perp(\hat{\theta})$, then*

$$\mathbb{E}[r^\|(\hat{\theta}_{t+1}) - r^\|(\hat{\theta}_{(t)})|\hat{\theta}_{(t)}] = -\frac{\alpha_{(t)}r^\|(\hat{\theta}_{(t)})}{(1-\eta)^2}\mathbf{v}_{(t)}^\top \left(\eta^2 D_{(t)} + I\right)\mathbf{v}_{(t)}\frac{\|X^\top \mathbf{1}\|_2^2}{n} + \mathcal{O}(\alpha_{(t)}^2 + \frac{1}{1-\eta})$$

$$\lesssim -\frac{\alpha_{(t)}r^\|(\hat{\theta}_{(t)})}{(1-\eta)^2}.$$

From this result we can conclude that this component of the residual converges very rapidly under the implicit parameterization. For instance, given $\alpha_t = \mathcal{O}(1)$, the expected convergence rate along the $\mathbf{1}$ direction is $\mathcal{O}(1/(1-\eta)^2)$, which is substantially faster than the more general rate, $\mathcal{O}(1/n)$, for the explicit and implicit case seen in Theorem 2 and 3, respectively. Moreover, this suggest that increasing $\eta$ will accentuate this effect.

Intuitively, this suggests that the implicit parameterization is favored when the residuals have similar magnitudes and sign, such as when every state is being over or underestimated. In practice, this is not uncommon since the estimates are often initialized to similar values, e.g., $\hat{V}(s) \approx 0$. Additionally, the true state values are typically highly correlated, making this systemic error all the more likely.

To further understand how the properties of the two parameterizations differ, we continue our analysis by examining how the variance of the SGD updates compares to the magnitude of their expectation which we quantify with the ratio of the variance and the squared mean update. If the variance is large compared to the expected updates, then the optimization is likely to spend a significant amount of time bouncing around, leading to slower convergence [2, 12, 25]. For this reason, it is desirable to have this ratio be as small as possible. We consider this variance-mean ratio instead of just the variance itself to avoid penalizing cases where the variance is large due to large expected improvements. In the rest of this section, we compare the performance of SGD with the explicit and implicit parameterization through this ratio which we state formally in the following theorem.

**Theorem 5.** *Given $n$ data points and a batch size of $k$, the (variance)/(mean square) ratio of SGD updates is as follows.*

*1. For the explicit parametrization,*

$$\frac{\mathbb{E}[\|\Delta\theta_{(t)}\|_2^2]}{\|\mathbb{E}[\Delta\theta_{(t)}]\|_2^2} = \frac{n(k-1)}{k(n-1)} + \frac{(n-k)n}{k(n-1)} \frac{\sum_i r_i^2 \|\phi(x_i)\|_2^2}{\|\mathbf{r}(\theta_{(t)})\|_2^2}.$$

*2. For the implicit parametrization,*

$$\frac{\mathbb{E}[\|\Delta\hat{\theta}_{(t)}\|_2^2]}{\|\mathbb{E}[\Delta\hat{\theta}_{(t)}]\|_2^2} = \frac{n(k-1)}{k(n-1)} + \frac{(n-k)n}{k(n-1)} \frac{\sum_i \langle \mathbf{1}, r_i\phi(x_i)\rangle^2}{(\sum_i \langle \mathbf{1}, r_i\phi(x_i)\rangle)^2} + \mathcal{O}(1-\eta).$$

Once more, we see the $\mathbf{1}$ direction play an important role in determining the behavior of the implicit case. Given the importance of the choice of features, $\phi$, it can be somewhat difficult to understand the implications of these results. To help build intuition, we conclude by examining the specials case when using tabular values where $\phi$ corresponds to a one-hot encoding with the following corollaries.

**Corollary 2.** *For the SGD with **explicit parametrization** and one-hot encoding the (variance)/(mean square) ratio of each noisy gradient update is given by*

$$\frac{\mathbb{E}[\|\Delta\theta_{(t)}\|_2^2]}{\|\mathbb{E}[\Delta\theta_{(t)}]\|_2^2} = \frac{n}{k}.$$

We see that under the explicit parametrization, the (variance)/(mean square) ratio is given by $n/k$, and is fixed throughout training. In contrast, Theorem 5 tells us that when using a one-hot encoding under the implicit parameterization, the dependency on the residual remains and will likely vary throughout the optimization. Similar to before, we can use these observations to help us better understand the conditions under which the implicit parameterization might exhibit preferable properties. The following corollary provides some insight as to what some of these conditions might be.

**Corollary 3.** *Assume the residuals are bounded such that $0 < r_{min} \leq [\mathbf{r}(\hat{\theta}_{(t)})]_i \leq r_{max}$ for all $1 \leq i \leq n$, then the (variance)/(mean square) ratio under the **implicit parameterization** is*

$$\frac{\mathbb{E}[\|\Delta\hat{\theta}_{(t)}\|_2^2]}{\|\mathbb{E}[\Delta\hat{\theta}_{(t)}]\|_2^2} \lesssim 1 + \frac{1}{k}\left(\frac{r_{max}}{r_{min}}\right)^2.$$

This result tells us that we should expect a considerably better (variance)/(mean square) ratio under the implicit parameterization when the ratio, $r_{max}/r_{min}$, of the largest and the smallest residual among the $n$ data points is small, as is the case when residuals are similar to each other. For instance, a "balanced" residuals such that $(r_{max}/r_{min})^2 = \mathcal{O}(1)$ means that the (variance)/(mean square) ratio of SGD updates is also $\mathcal{O}(1)$, whereas it would be $\mathcal{O}(n)$ under the explicit parameterization. This tells us that we can expect a notable benefit of using the implicit parameterization instead of the explicit one under balanced residuals. Finally, we note that this observation mirrors well the results from Theorems 2 and 4 which say that for a residual with a dominant $\mathbf{1}$ component, the convergence rate with the implicit parameterization is roughly $n$ times faster. These results strongly suggest that the implicit parameterization might be preferrable in situations where the residuals are expected to be highly correlated.

# 4 Empirical results

To illustrate our theoretical results, we evaluate the two parameterizations by fitting value functions from data collected *offline*. Specifically, we consider a setting where value targets are generated by rolling out a fixed stochastic policy and are thus noisy and imperfect similar to what might be encountered in applications of reinforcement learning. When evaluating, the residual is computed using the true values or, when dealing with a continuous state space, using unseen trajectories. All states in a trajectory are used with corresponding value targets, $y_i$, computed from the empirical discounted return. Consequently, duplicate states might be present in the dataset and might be given

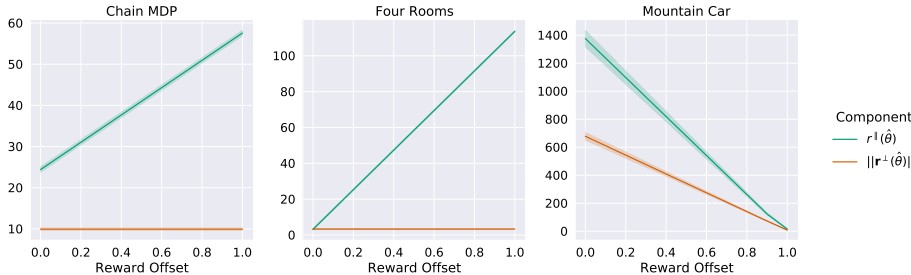

Figure 1: The effect of the reward offset on the components of the residual

different targets. All results show here except for Figure 2 consider the residual after 300 training steps. Since the purpose of these experiments are to illustrate our theoretical results, we leave the temporal difference learning case to future work in which we hope to extend our results.

To ensure any observed differences are caused by differing dynamics and not by differing initialization strategy, we use the same 20 seeds for both parameterizations and initialize parameters such that identical seeds result in the same initial linear weights, $\theta$ and $\hat{\theta}$. The shaded areas in our figures visualize the sample standard deviation across the seeds. For all experiments, we used a batch size $k = 25$. We ran these experiments with several combinations of learning rates and internal discounts but only present a few representative results here. Additional results and implementation details can be found in the appendix. All other implementation details, data and code are publicly available[1]. We consider three simple, illustrative domains: a chain MDP, the four rooms domain and the mountain car domain, which we describe below.

**Chain MDP:** a two-action MDP which requires an optimal agent to repeat the "correct" action until it has reached the end of the chain where the agent receives a large reward, $+10$, before being sent back to the start of the chain. At any point along the chain, the agent can execute the "wrong" action and collect a small reward, $+1$, before being prematurely sent back to the beginning. The chain MDP has 10 states along the good path and 1 extra transient state when executing the wrong action. States were represented with one-hot encoding and $\gamma = 0.9$ was used. Trajectories followed an $\epsilon$-greedy policy which picks a random action with probability $\epsilon = 0.1$ and an optimal action otherwise.

**Four rooms [21]:** a gridworld navigation task with obstacles dividing the world into 4 connected rooms. The agent can attempt to move in either of the four cardinal directions but will remain stationary with probability $1/3$. The goal is a terminal state with a single $+1$ reward situated at $(7, 1)$. As before, we use a one-hot encoding, an $\epsilon$-greedy policy with $\epsilon = 0.1$, and $\gamma = 0.9$.

**Mountain car [13, 20]:** a domain in which an agent must accumulate momentum by driving back and forth in order to get an underactuated car over a hill by applying three possible actions: idle, forward or backwards force. Each step results in a $-1$ reward except when transitioning to a terminal state. We encode the continuous state space using a uniformed 20x20 grid of normalized radial basis functions. Trajectories are sampled using an $\epsilon$-greedy energy pumping policy which picks a random action with probability $0.1$ and otherwise applies force in the direction of the car's velocity. Furthermore, in lieu of the true analytical values, we generate an additional test dataset and evaluate using similar but likely unseen states. In this domain, we use a discount factor $\gamma = 0.99$.

We consider a set of experiments that observe the change in the performance of both parameterizations as the rewards are offset by a constant. The purpose of these experiments is two fold: 1) to demonstrate the significance of the $\mathbf{1}$ component of the residual in a policy evaluation setting, and 2) to illustrate how increasing this component of the residual severely degrades the performance under the explicit case while having a lesser effect under the implicit parameterization as our theory would suggest.

The scale and skew of state values can have a significant impact on the learning performance. While these effects can be mitigated in a supervised learning setting, existing strategies aren't always applicable to a RL setting where values might not be known beforehand such as when using a temporal difference learning based approach. For this reason, it's not uncommon for values to be offset away from zero in unison making the $\mathbf{1}$ component more significant.

---

[1]https://github.com/gehring/implicit-estimators

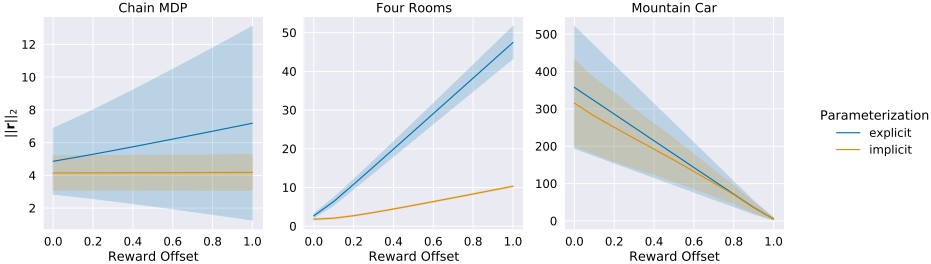

Figure 2: The effect of the reward offset on the norm of the final residual using $\eta = 0.95$

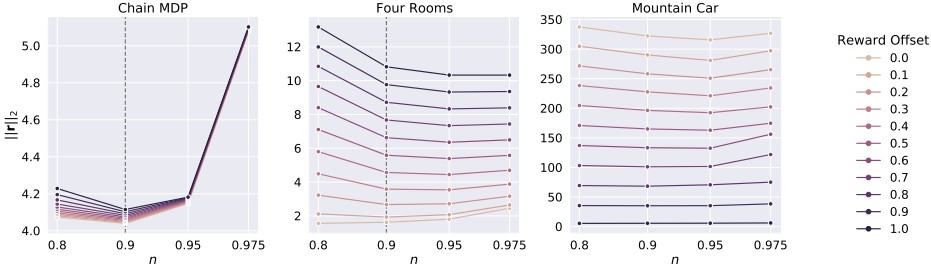

Figure 3: The effect of $\eta$ on the final residual under the implicit parameterization. The vertical dotted line represents the true discount factor used to generate values. Note that the x-axis uses a logit scale.

We illustrate this by adding a constant offset to the reward while keeping policies and transitions unchanged. In the case of mountain car, varying this offset from $0$ to $+1$ spans the two most common reward formulations for shortest-path like problems, the uniform step cost and the sparse reward approach where reaching the goal is the only non-zero reward. Figure 1 shows how the reward offset affects the residual by decomposing the residual at the start of training into two components: its projection on the $\mathbf{1}$ direction and the norm of the remainder. We see that the reward offset allows us to manipulate the relative magnitude of $\mathbf{1}$ component enabling us to illustrate our theory as discussed bellow.

We start by observing how changes in the $\mathbf{1}$ component affect each parameterizations. Figure 2 compares the residual at the end of training and show that, unsurprisingly, increasing the norm of the initial residual decreases the final performance. However, we also see a significant difference in the severity of this effect between the two cases. As established by Theorem 4, the implicit parameterization has a significant advantage in convergence rate of the residual's $\mathbf{1}$ component when compared to the explicit case. Thus, our theory predicts that the implicit parameterization would be less sensitive to changes in this component which is exactly what our empirical results show.

Finally, we conclude our empirical contributions by illustrating the role of $\eta$ in reducing the sensitivity to the $\mathbf{1}$ component of the residual. Figure 3 plots the final residual of each $\eta$ and compares between the different offsets. We can compare this sensitivity under different $\eta$'s by looking at the relative distance between the plots of a given reward offset. Note that, as is the case for the Chain MDP, the least sensitive $\eta$ isn't necessarily the best performing $\eta$. In addition, the best $\eta$ isn't necessarily equal to the true underlying discount factor, $\gamma$. This raises several interesting questions not covered by our theoretical results, such as how to select the "best" $\eta$, which we hope to explore in future work.

## 5   Conclusion

This work serves as a first step towards better understanding the properties of end-to-end model-based methods. This was done by first formulating a simple instance of such methods as an implicit parameterization of a linear function and analysing the dynamics of gradient descent under this parameterization. We showed that despite resulting in a non-linear and non-convex optimization problem, the implicitly parameterized linear weights still converges to a unique global optimal. Furthermore, we quantified the convergence rate of the residual under the implicit parameterization

showing a surprisingly fast convergence rate along the **1** direction. We then derive similar conclusions by deriving the ratio of the variance and squared mean SGD updates and identifying some conditions under which the implicit parameterization exhibits a significantly faster convergence rate than its explicit counterpart. We conclude our contributions with some empirical results which illustrate some of the phenomena identified in our theoretical results.

Although our analysis is restricted to a simple Markov reward chain model, we believe our results are likely to generalize to the full MDP setting since the subgradients of these two formulations have similar forms; both can be seen as backpropagating through a Markov reward chain. This is because the subderivatives of the Bellman optimality constraints only depend on the transitions under the optimal policy, i.e., the Markov reward chain induced by the optimal policy. We hope to extend our results to the more general MDP case in future work.

## 6 Acknowledgements

We gratefully acknowledge support from NSF grant 1723381; from AFOSR grant FA9550-17-1-0165; from ONR grant N00014-18-1-2847; from MIT-IBM Watson Lab; from the MIT Quest for Intelligence and from SUTD Temasek Laboratories.

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
