# A  Proofs of theoretical results

## A.1  Proof of Theorem 1

We first prove the following key lemma, which is used in the proof of Theorem 1.

**Lemma 1.** *Let* $P_{(t)} = A_{(t)}^{-1} \left( \eta^2 D_{(t)} + I \right) \left( A_{(t)}^{-1} \right)^\top$ *where* $D_{(t)} = \mathrm{diag}(\mathbf{d}_{(t)})$ *and* $\left[ \mathbf{d}_{(t)} \right]_k = \sum_i \left( \sum_j \frac{\partial \sigma(F_{(t)})_{ki}}{\partial \left[ F_{(t)} \right]_{kj}} \theta_j \right)^2$, *then* $\lambda_{\min}(P_{(t)}) \geq c$ *for some time-independent constant* $c > 0$, *and the following holds:*

$$\begin{aligned} \theta_{(t+1)} &= \theta_{(t)} - \alpha_{(t)} \nabla L(\theta_{(t)}), \\ \hat{\theta}_{(t+1)} &= \hat{\theta}_{(t)} - \alpha_{(t)} P_{(t)} \nabla L(\hat{\theta}_{(t)}) + \mathcal{O}(\alpha_{(t)}^2) \end{aligned} \tag{6}$$

*Proof of Lemma 1.* Recall that $L(q) = \frac{1}{2} \sum_i^n (\phi(x_i)^\top q - y_i)^2 = \frac{1}{2} \sum_i^n r_i(q)^2$ for any $q \in \mathbb{R}^k$ where $r_i(q) = (\phi(x_i)^\top q - y_i)$. We define

$$L_i(q) := \frac{1}{2} (\phi(x_i)^\top q - y_i)^2 = \frac{1}{2} r_i(q)^2, \tag{7}$$

for which we have that for any $q \in \mathbb{R}^k$,

$$\nabla L_i(q) = r_i(q) \phi(x_i) \in \mathbb{R}^k. \tag{8}$$

Note that

$$L(q) = \sum_i^n L_i(q) \tag{9}$$

and

$$\nabla L(q) = \sum_i^n \nabla L_i(q). \tag{10}$$

By viewing the $\hat{\theta} = (I - \eta\sigma(F))^{-1} w$ as the function of $F$ and $w$, we define

$$\nabla_F L_i(\hat{\theta}) := \frac{\partial L_i(\hat{\theta})}{\partial F} \quad \text{and} \quad \nabla_w L_i(\hat{\theta}) := \frac{\partial L_i(\hat{\theta})}{\partial w}, \tag{11}$$

where

$$L_i(\hat{\theta}) = \frac{1}{2} (\phi(x_i)^\top \hat{\theta} - y_i)^2 = \frac{1}{2} (\phi(x_i)^\top (I - \eta\sigma(F))^{-1} w - y_i)^2. \tag{12}$$

For the explicit parameterization, by the definition of the gradient dynamics,

$$\theta_{(t+1)} = \theta_{(t)} - \alpha_{(t)} \nabla L(q)(\theta). \tag{13}$$

Therefore, in the rest of the proof, we analyze the dynamics of the implicit parameterization:

$$F_{(t+1)} = F_{(t)} - \alpha_t \nabla_F L(\hat{\theta}_{(t)}) \tag{14}$$

$$w_{(t+1)} = w_{(t)} - \alpha_t \nabla_w L(\hat{\theta}_{(t)}), \tag{15}$$

which induces the dynamics of

$$\hat{\theta}_{(t+1)} = [I - \eta(F_{(t)} - \alpha_t \nabla_F L_{i_t}(\hat{\theta}_{(t)}))]^{-1} [w_{(t)} - \alpha_t \nabla_w L_{i_t}(\hat{\theta}_{(t)})]. \tag{16}$$

Recall that $A = I - \eta\sigma(F)$. Using the chain rule, we have that for any differentiable function $g : A \mapsto g(A)$,

$$\frac{\partial g(A)}{\partial F_{kl}} = \sum_{i=1}^n \sum_{j=1}^n \frac{\partial g(A)}{\partial A_{ij}} \frac{\partial A_{ij}}{\partial F_{kl}} = \sum_{i=1}^n \sum_{j=1}^n \frac{\partial g(A)}{\partial A_{ij}} \frac{\partial A_{ij}}{\partial \sigma(F)_{ij}} \frac{\partial \sigma(F)_{ij}}{\partial F_{kl}} = -\eta \sum_{i=1}^n \sum_{j=1}^n \frac{\partial g(A)}{\partial A_{ij}} \frac{\partial \sigma(F)_{ij}}{\partial F_{kl}}. \tag{17}$$

Here, we calculate the term $\frac{\partial \sigma(F)_{ij}}{\partial F_{kl}}$ using the quotient rule as follows:

$$\frac{\partial \sigma(F)_{ij}}{\partial F_{kl}} = \frac{\partial}{\partial F_{kl}} \frac{\exp(F_{ij})}{\sum_t \exp(F_{it})} \tag{18}$$

$$= \frac{(\frac{\partial \exp(F_{ij})}{\partial F_{kl}})(\sum_t \exp(F_{it})) - \exp(F_{ij})(\frac{\partial \sum_t \exp(F_{it})}{\partial F_{kl}})}{(\sum_t \exp(F_{it}))^2} \tag{19}$$

$$= \frac{\mathbb{1}\{i=k\}\mathbb{1}\{j=l\} \exp(F_{ij})(\sum_t \exp(F_{it})) - \mathbb{1}\{i=k\} \exp(F_{ij}) \exp(F_{il})}{(\sum_t \exp(F_{it}))^2} \tag{20}$$

$$= \frac{\mathbb{1}\{i=k\}\mathbb{1}\{j=l\} \exp(F_{ij})}{\sum_t \exp(F_{it})} - \frac{\mathbb{1}\{i=k\} \exp(F_{ij}) \exp(F_{il})}{(\sum_t \exp(F_{it}))^2} \tag{21}$$

$$= \mathbb{1}\{i=k\}\mathbb{1}\{j=l\}\sigma(F)_{ij} - \mathbb{1}\{i=k\}\frac{\exp(F_{ij})}{\sum_t \exp(F_{it})}\frac{\exp(F_{il})}{\sum_t \exp(F_{it})} \tag{22}$$

$$= \mathbb{1}\{i=k\}\mathbb{1}\{j=l\}\sigma(F)_{ij} - \mathbb{1}\{i=k\}\sigma(F)_{ij}\sigma(F)_{il} \tag{23}$$

Combining the two equations above,

$$\frac{\partial g(A)}{\partial F_{kl}} = -\eta \sum_{j=1}^{n} \frac{\partial g(A)}{\partial A_{kj}} \frac{\partial \sigma(F)_{kj}}{\partial F_{kl}} = -\eta \frac{\partial g(A)}{\partial A_{k\bullet}} \frac{\partial \sigma(F)_{k\bullet}}{\partial F_{kl}} \in \mathbb{R}, \tag{24}$$

where $\frac{\partial g(A)}{\partial A_{k\bullet}} \in \mathbb{R}^{1\times n}$ and $\frac{\partial \sigma(F)_{k\bullet}}{\partial F_{kl}} \in \mathbb{R}^{n\times 1}$. This yields

$$\frac{\partial g(A)}{\partial F_{k\bullet}} = -\eta \frac{\partial g(A)}{\partial A_{k\bullet}} \frac{\partial \sigma(F)_{k\bullet}}{\partial F_{k\bullet}} \in \mathbb{R}^{1\times n}, \tag{25}$$

where $\frac{\partial \sigma(F)_{k\bullet}}{\partial F_{k\bullet}} \in \mathbb{R}^{n\times n}$. By setting $g(A) = \phi(x_i)^\top \hat{\theta}$,

$$\frac{\partial g(A)}{\partial F} = \frac{\partial \phi(x_i)^\top \hat{\theta}}{\partial F} = -\eta \begin{bmatrix} \frac{\partial \phi(x_i)^\top \hat{\theta}}{\partial A_{1\bullet}} \frac{\partial \sigma(F)_{1\bullet}}{\partial F_{1\bullet}} \\ \vdots \\ \frac{\partial \phi(x_i)^\top \hat{\theta}}{\partial A_{n\bullet}} \frac{\partial \sigma(F)_{n\bullet}}{\partial F_{n\bullet}} \end{bmatrix}. \tag{26}$$

Using $\frac{\partial a^\top X^{-1} b}{\partial X} = -X^{-\top} a b^\top X^{-\top}$, we have that

$$\frac{\partial \phi(x_i)^\top \hat{\theta}}{\partial A} = \frac{\partial \phi(x_i)^\top A^{-1} w}{\partial A} = -A^{-\top} \phi(x_i) w^\top A^{-\top}, \tag{27}$$

which yields

$$\frac{\partial \phi(x_i)^\top \hat{\theta}}{\partial A_{k\bullet}} = -(A^{-\top} \phi(x_i) w^\top A^{-\top})_{k\bullet} = -(A^{-\top} \phi(x_i))_k w^\top A^{-\top} = -(A^{-\top})_{k\bullet} \phi(x_i) w^\top A^{-\top}. \tag{28}$$

Combining (26) and (28),

$$\frac{\partial \phi(x_i)^\top \hat{\theta}}{\partial F} = -\eta \begin{bmatrix} -(A^{-\top})_{1\bullet}\phi(x_i)w^\top A^{-\top}\frac{\partial \sigma(F)_{1\bullet}}{\partial F_{1\bullet}} \\ \vdots \\ -(A^{-\top})_{n\bullet}\phi(x_i)w^\top A^{-\top}\frac{\partial \sigma(F)_{n\bullet}}{\partial F_{n\bullet}} \end{bmatrix} = \eta \begin{bmatrix} (A^{-\top})_{1\bullet}\phi(x_i)w^\top A^{-\top}\frac{\partial \sigma(F)_{1\bullet}}{\partial F_{1\bullet}} \\ \vdots \\ (A^{-\top})_{n\bullet}\phi(x_i)w^\top A^{-\top}\frac{\partial \sigma(F)_{n\bullet}}{\partial F_{n\bullet}} \end{bmatrix} \tag{29}$$

$$= \eta \begin{bmatrix} (A^{-\top})_{1\bullet}\phi(x_i)\hat{\theta}^\top\frac{\partial \sigma(F)_{1\bullet}}{\partial F_{1\bullet}} \\ \vdots \\ (A^{-\top})_{n\bullet}\phi(x_i)\hat{\theta}^\top\frac{\partial \sigma(F)_{n\bullet}}{\partial F_{n\bullet}} \end{bmatrix}, \tag{30}$$

where the last line follows from $\hat{\theta} = A^{-1}w$. Using chain rule with $L_i(\hat{\theta}) := \frac{1}{2}(\phi(x_i)^\top \hat{\theta} - y_i)^2$,

$$\nabla_F L_i(\hat{\theta}) = \frac{\partial L_i(\hat{\theta})}{\partial F} = (\phi(x_i)^\top \hat{\theta} - y_i)\frac{\partial \phi(x_i)^\top \hat{\theta}}{\partial F} \tag{31}$$

$$= \eta(\phi(x_i)^\top \hat{\theta} - y_i) \begin{bmatrix} (A^{-\top})_{1\bullet}\phi(x_i)\hat{\theta}^\top \frac{\partial\sigma(F)_{1\bullet}}{\partial F_{1\bullet}} \\ \vdots \\ (A^{-\top})_{n\bullet}\phi(x_i)\hat{\theta}^\top \frac{\partial\sigma(F)_{n\bullet}}{\partial F_{n\bullet}} \end{bmatrix}. \tag{32}$$

By defining $G_k = \frac{\partial\sigma(F)_{k\bullet}}{\partial F_{k\bullet}} \in \mathbb{R}^{n\times n}$,

$$\nabla_F L_i(\hat{\theta})_{k\bullet} = \eta(\phi(x_i)^\top \hat{\theta} - y_i)(A^{-\top})_{k\bullet}\phi(x_i)\hat{\theta}^\top G_k \in \mathbb{R}^{1\times n}. \tag{33}$$

Let $\mathcal{I}$ be an arbitrary subset of $\{1,\ldots,n\}$. Then, by setting $\varphi_{ij}(\alpha) = \sigma(F - \alpha\sum_{q\in\mathcal{I}}\nabla_F L_q(\hat{\theta}))_{ij} \in \mathbb{R}$,

$$\sigma(F - \alpha\sum_{q\in\mathcal{I}}\nabla_F L_q(\hat{\theta}))_{ij} = \varphi_{ij}(\alpha) = \varphi_{ij}(0) + \frac{\partial\varphi_{ij}(0)}{\partial\alpha}\alpha + o(\alpha^2). \tag{34}$$

Using the chain rule $\frac{\partial g(A)}{\partial x} = \sum_i \sum_j \frac{\partial g(A)}{\partial A_{ij}}\frac{\partial A_{ij}}{\partial x}$ (for any differentiable function $g$), and by setting $M = F - \alpha\sum_{q\in\mathcal{I}}\nabla_F L_q(\hat{\theta}) \in \mathbb{R}^{n\times n}$,

$$\frac{\partial\varphi_{ij}(\alpha)}{\partial\alpha} = \sum_{k=1}^n \sum_{l=1}^n \frac{\partial\sigma(M)_{ij}}{\partial M_{kl}}\frac{\partial M_{kl}}{\partial\alpha} \tag{35}$$

$$= -\sum_{k=1}^n \sum_{l=1}^n [\mathbb{1}\{i=k\}\mathbb{1}\{j=l\}\sigma(M)_{ij} - \mathbb{1}\{i=k\}\sigma(M)_{ij}\sigma(M)_{il}] \sum_{q\in\mathcal{I}}\nabla_F L_q(\hat{\theta})_{kl} \tag{36}$$

$$= -\sum_{l=1}^n [\mathbb{1}\{j=l\}\sigma(M)_{ij} - \sigma(M)_{ij}\sigma(M)_{il}] \sum_{q\in\mathcal{I}}\nabla_F L_q(\hat{\theta})_{il} \tag{37}$$

$$= -\frac{\partial\sigma(M)_{ij}}{\partial M_{i\bullet}}\left(\sum_{q\in\mathcal{I}}\nabla_F L_q(\hat{\theta})_{i\bullet}\right)^\top \in \mathbb{R} \tag{38}$$

$$= -\sum_{q\in\mathcal{I}}\nabla_F L_q(\hat{\theta})_{i\bullet}\left(\frac{\partial\sigma(M)_{ij}}{\partial M_{i\bullet}}\right)^\top \tag{39}$$

Therefore,

$$\frac{\partial\varphi_{ij}(0)}{\partial\alpha} = -\sum_{q\in\mathcal{I}}\nabla_F L_q(\hat{\theta})_{i\bullet}\left(\frac{\partial\sigma(F)_{ij}}{\partial F_{i\bullet}}\right)^\top, \tag{40}$$

which implies that

$$\frac{\partial\varphi_{k\bullet}(0)}{\partial\alpha} = -\sum_{q\in\mathcal{I}}\nabla_F L_q(\hat{\theta})_{k\bullet}\left(\frac{\partial\sigma(F)_{k\bullet}}{\partial F_{k\bullet}}\right)^\top = -\sum_{q\in\mathcal{I}}\nabla_F L_q(\hat{\theta})_{k\bullet}G_k^\top \in \mathbb{R}^{1\times n}. \tag{41}$$

Summarizing these yields that

$$\frac{\partial\varphi(0)}{\partial\alpha} = -\begin{bmatrix} \sum_{q\in\mathcal{I}}\nabla_F L_q(\hat{\theta})_{1\bullet}G_1^\top \\ \vdots \\ \sum_{q\in\mathcal{I}}\nabla_F L_q(\hat{\theta})_{n\bullet}G_n^\top \end{bmatrix} \in \mathbb{R}^{n\times n} \tag{42}$$

By combining (34) and (41),

$$\sigma(F - \alpha\sum_{q\in\mathcal{I}}\nabla_F L_q(\hat{\theta})) = \varphi(0) + \alpha\frac{\partial\varphi(0)}{\partial\alpha} + o(\alpha^2)$$

$$= \sigma(F) - \alpha\begin{bmatrix} \sum_{q\in\mathcal{I}}\nabla_F L_q(\hat{\theta})_{1\bullet}G_1^\top \\ \vdots \\ \sum_{q\in\mathcal{I}}\nabla_F L_q(\hat{\theta})_{n\bullet}G_n^\top \end{bmatrix} + o(\alpha^2). \tag{43}$$

Using (43),

$$[I - \eta\sigma(F - \alpha\sum_{q\in\mathcal{I}}\nabla_F L_q(F))]^{-1} = \left[I - \eta\sigma(F) + \alpha\eta\begin{bmatrix}\sum_{q\in\mathcal{I}}\nabla_F L_q(\hat\theta)_{1\bullet}G_1^\top\\ \vdots\\ \sum_{q\in\mathcal{I}}\nabla_F L_q(\hat\theta)_{n\bullet}G_n^\top\end{bmatrix} + o(\alpha^2)\right]^{-1} \tag{44}$$

$$= \left[A + \alpha\eta\begin{bmatrix}\sum_{q\in\mathcal{I}}\nabla_F L_q(\hat\theta)_{1\bullet}G_1^\top\\ \vdots\\ \sum_{q\in\mathcal{I}}\nabla_F L_q(\hat\theta)_{n\bullet}G_n^\top\end{bmatrix} + o(\alpha^2)\right]^{-1} \tag{45}$$

By setting $M = \begin{bmatrix}\sum_{q\in\mathcal{I}}\nabla_F L_q(\hat\theta)_{1\bullet}G_1^\top\\ \vdots\\ \sum_{q\in\mathcal{I}}\nabla_F L_q(\hat\theta)_{n\bullet}G_n^\top\end{bmatrix}$ and $\varphi(\alpha) = [A + \alpha\eta M + o(\alpha^2)]^{-1}$ and by using

$\frac{\partial Y^{-1}}{\partial x} = -Y^{-1}\frac{\partial Y}{\partial x}Y^{-1}$,

$$[I - \eta\sigma(F - \alpha\sum_{q\in\mathcal{I}}\nabla L_q(F))]^{-1} = [A + \alpha\eta M + o(\alpha^2)]^{-1} \tag{46}$$

$$= \varphi(\alpha) \tag{47}$$

$$= \varphi(0) + \frac{\partial\varphi(0)}{\partial\alpha}\alpha + o(\alpha^2) \tag{48}$$

$$= A^{-1} - \alpha\eta A^{-1}MA^{-1} + 2\alpha o(\alpha) + o(\alpha^2) \tag{49}$$

$$= A^{-1} - \alpha\eta A^{-1}MA^{-1} + o(\alpha^2) \tag{50}$$

Then, using this and (33),

$$[I - \eta\sigma(F - \alpha\sum_{i\in\mathcal{I}}\nabla L_i(F))]^{-1} \tag{51}$$

$$= A^{-1} - \alpha\eta A^{-1}\begin{bmatrix}\sum_{i\in\mathcal{I}}\nabla_F L_i(\hat\theta)_{1\bullet}G_1^\top\\ \vdots\\ \sum_{i\in\mathcal{I}}\nabla_F L_i(\hat\theta)_{n\bullet}G_n^\top\end{bmatrix}A^{-1} + o(\alpha^2) \tag{52}$$

$$= A^{-1} - \alpha\eta\sum_{i\in\mathcal{I}}A^{-1}\begin{bmatrix}\eta(\phi(x_i)^\top\hat\theta - y_i)(A^{-\top})_{1\bullet}\phi(x_i)\hat\theta^\top G_1 G_1^\top\\ \vdots\\ \eta(\phi(x_i)^\top\hat\theta - y_i)(A^{-\top})_{n\bullet}\phi(x_i)\hat\theta^\top G_n G_n^\top\end{bmatrix}A^{-1} + o(\alpha^2) \tag{53}$$

$$= A^{-1} - \alpha\eta^2\sum_{i\in\mathcal{I}}(\phi(x_i)^\top\hat\theta - y_i)A^{-1}\begin{bmatrix}(A^{-\top})_{1\bullet}\phi(x_i)\hat\theta^\top G_1 G_1^\top\\ \vdots\\ (A^{-\top})_{n\bullet}\phi(x_i)\hat\theta^\top G_n G_n^\top\end{bmatrix}A^{-1} + o(\alpha^2). \tag{54}$$

For the gradient with respect to $w$, we have

$$\nabla_w L_i(\hat\theta) = \nabla_w\frac{1}{2}(\phi(x_i)^\top A^{-1}w - y_i)^2 = A^{-\top}(r_i(\theta)\phi(x_i)) = A^{-\top}\nabla L_i(\theta_{(t)}). \tag{55}$$

Therefore, using $\hat\theta_{(t)} = A_{(t)}^{-1}w_{(t)}$ with $\mathcal{I}_t$ being an arbitrary subset of $\{1,\ldots,n\}$,

$$\hat\theta_{(t+1)} \tag{56}$$

$$= [I - \eta(F_{(t)} - \alpha_t\sum_{i\in\mathcal{I}_t}\nabla_F L_i(\hat\theta_{(t)}))]^{-1}[w_{(t)} - \alpha_t\sum_{i\in\mathcal{I}_t}\nabla_w L_i(\hat\theta_{(t)})] \tag{57}$$

$$= \left[A^{-1} - \alpha_t\eta^2\sum_{i\in\mathcal{I}_t}r_i(\hat\theta_{(t)})A_{(t)}^{-1}\begin{bmatrix}(A_{(t)}^{-\top})_{1\bullet}\phi(x_i)\hat\theta_{(t)}^\top G_1 G_1^\top\\ \vdots\\ (A_{(t)}^{-\top})_{n\bullet}\phi(x_i)\hat\theta_{(t)}^\top G_n G_n^\top\end{bmatrix}A_{(t)}^{-1} + o(\alpha_t^2)\right][w_{(t)} - \alpha_t\sum_{i\in\mathcal{I}_t}A_{(t)}^{-\top}\nabla L_i(\hat\theta_{(t)})]$$

$$\tag{58}$$

$$= A_{(t)}^{-1} w_{(t)} - \alpha_t \eta^2 \sum_{i \in \mathcal{I}_t} r_i(\hat{\theta}_{(t)}) A_{(t)}^{-1} \begin{bmatrix} (A_{(t)}^{-\top})_{1\bullet} \phi(x_i) \hat{\theta}_{(t)}^{\top} G_1 G_1^{\top} \\ \vdots \\ (A_{(t)}^{-\top})_{n\bullet} \phi(x_i) \hat{\theta}_{(t)}^{\top} G_n G_n^{\top} \end{bmatrix} A_{(t)}^{-1} w_{(t)} - \alpha_t A_{(t)}^{-1} A_{(t)}^{-\top} \sum_{i \in \mathcal{I}_t} \nabla L_i(\hat{\theta}_{(t)}) + o(\alpha_t^2)$$

$$\tag{59}$$

$$= \hat{\theta}_{(t)} - \alpha_t \eta \sum_{i \in \mathcal{I}_t} r_i(\hat{\theta}_{(t)}) A_{(t)}^{-1} \begin{bmatrix} (A_{(t)}^{-\top})_{1\bullet} \phi(x_i) \hat{\theta}_{(t)}^{\top} G_1 G_1^{\top} \\ \vdots \\ (A_{(t)}^{-\top})_{n\bullet} \phi(x_i) \hat{\theta}_{(t)}^{\top} G_n G_n^{\top} \end{bmatrix} \hat{\theta}_{(t)} - \alpha_t A_{(t)}^{-1} A_{(t)}^{-\top} \sum_{i \in \mathcal{I}_t} \nabla L_i(\hat{\theta}_{(t)}) + o(\alpha_t^2)$$

$$\tag{60}$$

$$= \hat{\theta}_{(t)} - \alpha_t \eta^2 \sum_{i \in \mathcal{I}_t} r_i(\hat{\theta}_{(t)}) A_{(t)}^{-1} \begin{bmatrix} (A_{(t)}^{-\top})_{1\bullet} \phi(x_i) \hat{\theta}_{(t)}^{\top} G_1 G_1^{\top} \hat{\theta}_{(t)} \\ \vdots \\ (A_{(t)}^{-\top})_{n\bullet} \phi(x_i) \hat{\theta}_{(t)}^{\top} G_n G_n^{\top} \hat{\theta}_{(t)} \end{bmatrix} - \alpha_t A_{(t)}^{-1} A_{(t)}^{-\top} \sum_{i \in \mathcal{I}_t} \nabla L_i(\hat{\theta}_{(t)}) + o(\alpha_t^2)$$

$$\tag{61}$$

$$= \hat{\theta}_{(t)} - \alpha_t \eta \sum_{i \in \mathcal{I}_t} r_i(\hat{\theta}_{(t)}) A_{(t)}^{-1} \mathrm{diag} \left( \begin{bmatrix} \hat{\theta}_{(t)}^{\top} G_1 G_1^{\top} \hat{\theta}_{(t)} \\ \vdots \\ \hat{\theta}_{(t)}^{\top} G_n G_n^{\top} \hat{\theta}_{(t)} \end{bmatrix} \right) \begin{bmatrix} (A_{(t)}^{-\top})_{1\bullet} \phi(x_i) \\ \vdots \\ (A_{(t)}^{-\top})_{n\bullet} \phi(x_i) \end{bmatrix} - \alpha_t A_{(t)}^{-1} A_{(t)}^{-\top} \sum_{i \in \mathcal{I}_t} \nabla L_i(\hat{\theta}_{(t)}) + o(\alpha_t^2)$$

$$\tag{62}$$

$$= \hat{\theta}_{(t)} - \alpha_t \eta^2 \sum_{i \in \mathcal{I}_t} r_i(\hat{\theta}_{(t)}) A_{(t)}^{-1} \mathrm{diag} \left( \begin{bmatrix} \hat{\theta}_{(t)}^{\top} G_1 G_1^{\top} \hat{\theta}_{(t)} \\ \vdots \\ \hat{\theta}_{(t)}^{\top} G_n G_n^{\top} \hat{\theta}_{(t)} \end{bmatrix} \right) \begin{bmatrix} (A_{(t)}^{-\top})_{1\bullet} \\ \vdots \\ (A_{(t)}^{-\top})_{n\bullet} \end{bmatrix} \phi(x_i) - \alpha_t A_{(t)}^{-1} A_{(t)}^{-\top} \sum_{i \in \mathcal{I}_t} \nabla L_i(\hat{\theta}_{(t)}) + o(\alpha_t^2)$$

$$\tag{63}$$

$$= \hat{\theta}_{(t)} - \alpha_t \eta^2 A_{(t)}^{-1} \mathrm{diag} \left( \begin{bmatrix} \hat{\theta}_{(t)}^{\top} G_1 G_1^{\top} \hat{\theta}_{(t)} \\ \vdots \\ \hat{\theta}_{(t)}^{\top} G_n G_n^{\top} \hat{\theta}_{(t)} \end{bmatrix} \right) A_{(t)}^{-\top} \sum_{i \in \mathcal{I}_t} (r_i(\hat{\theta}_{(t)}) \phi(x_{i_t})) - \alpha_t A_{(t)}^{-1} A_{(t)}^{-\top} \sum_{i \in \mathcal{I}_t} \nabla L_i(\hat{\theta}_{(t)}) + o(\alpha_t^2)$$

$$\tag{64}$$

$$= \hat{\theta}_{(t)} - \alpha_t \eta^2 A_{(t)}^{-1} \mathrm{diag} \left( \begin{bmatrix} \hat{\theta}_{(t)}^{\top} G_1 G_1^{\top} \hat{\theta}_{(t)} \\ \vdots \\ \hat{\theta}_{(t)}^{\top} G_n G_n^{\top} \hat{\theta}_{(t)} \end{bmatrix} \right) A_{(t)}^{-\top} \sum_{i \in \mathcal{I}_t} \nabla L_i(\hat{\theta}_{(t)}) - \alpha_t A_{(t)}^{-1} A_{(t)}^{-\top} \sum_{i \in \mathcal{I}_t} \nabla L_i(\hat{\theta}_{(t)}) + o(\alpha_t^2)$$

$$\tag{65}$$

$$= \hat{\theta}_{(t)} - \alpha_t \left( \eta^2 A_{(t)}^{-1} \mathrm{diag} \left( \begin{bmatrix} \hat{\theta}_{(t)}^{\top} G_1 G_1^{\top} \hat{\theta}_{(t)} \\ \vdots \\ \hat{\theta}_{(t)}^{\top} G_n G_n^{\top} \hat{\theta}_{(t)} \end{bmatrix} \right) A_{(t)}^{-\top} + A_{(t)}^{-1} A_{(t)}^{-\top} \right) \sum_{i \in \mathcal{I}_t} \nabla L_i(\hat{\theta}_{(t)}) + o(\alpha_t^2)$$

$$\tag{66}$$

$$= \hat{\theta}_{(t)} - \alpha_t \left( \eta^2 A_{(t)}^{-1} \mathrm{diag} \left( \begin{bmatrix} \|G_1^{\top} \hat{\theta}_{(t)}\|_2^2 \\ \vdots \\ \|G_n^{\top} \hat{\theta}_{(t)}\|_2^2 \end{bmatrix} \right) A_{(t)}^{-\top} + A_{(t)}^{-1} A_{(t)}^{-\top} \right) \sum_{i \in \mathcal{I}_t} \nabla L_i(\hat{\theta}_{(t)}) + o(\alpha_t^2) \tag{67}$$

$$= \hat{\theta}_{(t)} - \alpha_t P_{(t)} \sum_{i \in \mathcal{I}_t} \nabla L_i(\hat{\theta}_{(t)}) + o(\alpha_t^2), \tag{68}$$

where

$$P_{(t)} = \left( \eta^2 A_{(t)}^{-1} \mathrm{diag} \left( \begin{bmatrix} \|G_1^{\top} \hat{\theta}_{(t)}\|_2^2 \\ \vdots \\ \|G_n^{\top} \hat{\theta}_{(t)}\|_2^2 \end{bmatrix} \right) A_{(t)}^{-\top} + A_{(t)}^{-1} A_{(t)}^{-\top} \right). \tag{69}$$

Since $\mathcal{I}_t$ is an arbitrary subset of $\{1, \ldots, n\}$, we can set $\mathcal{I} = \{1, \ldots, n\}$ which yields that

$$\hat{\theta}_{(t+1)} = \hat{\theta}_{(t)} - \alpha_t P_{(t)} \nabla L(\hat{\theta}_{(t)}) + o(\alpha_t^2). \tag{70}$$

This along with equation (13) proves the statement of this theorem for the equation of the dynamics. In the following, we prove that $P_{(t)}$ is positive definite. Since $\|G_k^\top \hat{\theta}_{(t)}\|_2^2 \geq 0$ for all $k$, we can decompose

$$D := \text{diag}\left(\begin{bmatrix} \|G_1^\top \hat{\theta}_{(t)}\|_2^2 \\ \vdots \\ \|G_n^\top \hat{\theta}_{(t)}\|_2^2 \end{bmatrix}\right) = D^{1/2} D^{1/2}, \tag{71}$$

where

$$D^{1/2} = \text{diag}\left(\begin{bmatrix} \sqrt{\|G_1^\top \hat{\theta}_{(t)}\|_2^2} \\ \vdots \\ \sqrt{\|G_n^\top \hat{\theta}_{(t)}\|_2^2} \end{bmatrix}\right). \tag{72}$$

Thus,

$$\eta^2 A^{-1} \text{diag}\left(\begin{bmatrix} \|G_1^\top \hat{\theta}_{(t)}\|_2^2 \\ \vdots \\ \|G_n^\top \hat{\theta}_{(t)}\|_2^2 \end{bmatrix}\right) A^{-\top} = \eta^2 A^{-1} D^{1/2} D^{1/2} A^{-\top} = \eta^2 (A^{-1} D^{1/2})(A^{-1} D^{1/2})^\top \succeq 0 \tag{73}$$

Since a positive semidefinite matrix plus a positive definite matrix is positive definite (which directly follows from the definitions), and since $A_{(t)}^{-1} A_{(t)}^{-\top}$ is positive definite,

$$P_{(t)} = \left(\eta^2 A_{(t)}^{-1} \text{diag}\left(\begin{bmatrix} \|G_1^\top \hat{\theta}_{(t)}\|_2^2 \\ \vdots \\ \|G_n^\top \hat{\theta}_{(t)}\|_2^2 \end{bmatrix}\right) A_{(t)}^{-\top} + A_{(t)}^{-1} A_{(t)}^{-\top}\right) \succ 0. \tag{74}$$

Moreover, we notice that $A_{(t)}^{-1} A_{(t)}^{-\top}$ in $P_{(t)}$ is always positive definite for all $t$ and when $t \to \infty$ because $\|F\|_2^2 \to \infty$ or $\|F\|_2^2 \to 0$ do not cause $A_{(t)}^{-1} A_{(t)}^{-\top}$ to become rank deficient due to the definition of the matrix $A$ with the function $\sigma$. Therefore, $\lambda_{\min}(P_{(t)}) > 0$ for all $t$ and $\liminf_{t \to \infty} \lambda_{\min}(P_{(t)}) > 0$. □

Using Lemma 1, we prove Theorem 1 in the following:

*Proof of Theorem 1.* Lemma 1 shows that

$$\hat{\theta}_{(t+1)} = \hat{\theta}_{(t)} - \alpha_{(t)} P_{(t)} \nabla L(\hat{\theta}_{(t)}) + \mathcal{O}(\alpha_{(t)}^2),$$

where $P_{(t)} \succ 0$ for all $t$. This implies that

$$\frac{\hat{\theta}_{(t+1)} - \hat{\theta}_{(t)}}{\alpha_{(t)}} = -P_{(t)} \nabla L(\hat{\theta}_{(t)}) + O(\alpha_{(t)}),$$

where $\alpha_{(t)} > 0$. By recalling the definition of the Euler method and defining $\hat{\theta}(t) = \hat{\theta}_{(t)}$, we can rewrite this as

$$\frac{\hat{\theta}(t + \alpha) - \hat{\theta}(t)}{\alpha} = -P_{(t)} \nabla L(\hat{\theta}_{(t)}) + O(\alpha_{(t)}).$$

By taking the limit for $\alpha \to 0$ and going back to continuous-time dynamics, this implies that

$$\frac{d}{dt} \hat{\theta}_{(t)} = -P_{(t)} \nabla L(\hat{\theta}_{(t)}). \tag{75}$$

□

## A.2 Proof of Corollary 1

**Corollary 1.** *For any initialization $\hat{\theta}_{(0)}$, the gradient dynamics $\frac{d}{dt}\hat{\theta}_{(t)}$ converges to a global minimum.*

*Proof.* Theorem 1 shows that

$$\frac{d\hat{\theta}_{(t)}}{dt} = -P_{(t)}\nabla L(\hat{\theta}_{(t)}),$$

where $\lambda_{\min}(P_{(t)}) \geq c$ for some time-independent constant $c > 0$. By taking derivative of $L(\hat{\theta}_{(t)}) - L(\hat{\theta}^*)$ with respect to time $t$,

$$\frac{d}{dt}L(\hat{\theta}_{(t)}) = \frac{dL(\hat{\theta}_{(t)})}{d\hat{\theta}_{(t)}}\frac{d\hat{\theta}_{(t)}}{dt} = -\nabla L(\hat{\theta}_{(t)})^\top P_{(t)}\nabla L(\hat{\theta}_{(t)})$$

$$\leq -\lambda_{\min}(P_{(t)})\|\nabla L(\hat{\theta}_{(t)})\|_2^2$$

$$\leq -c\|\nabla L(\hat{\theta}_{(t)})\|_2^2,$$

where we used the chain rule. Since $c > 0$, this shows that the loss value always decreases unless $\|\nabla L(\hat{\theta}_{(t)})\|_2^2 = 0$. Since $L(\hat{\theta}^*) \geq 0$, this implies that $\|\nabla L(\hat{\theta}_{(t)})\|_2^2 \to 0$ as $t \to \infty$ under the gradient flow. Since $\|\nabla L(\hat{\theta}_{(t)})\|_2^2 = 0$ implies that $\nabla L(\hat{\theta}_{(t)}) = 0$, this means that $\hat{\theta}_{(t)}$ converges to a stationary point of $L$. Since the function $q \mapsto L(q)$ is convex and every stationary point $L$ is a global minimum, this implies the convergence to a global minimum. $\qquad\square$

## A.3 Proof of Theorem 2

**Theorem 2.** *Given a sequence of learning rates $(\alpha_{(t)})_t$, for SGD with the explicit parameterization, the expected decrease of the loss function satisfies*

$$\mathbb{E}[(L(\theta_{t+1}) - L(\theta_{(t)}))|\theta_{(t)}] = -\frac{\alpha_{(t)}}{n}\left\|\mathbf{r}(\theta_{(t)})\right\|_2^2 + \mathcal{O}(\alpha_{(t)}^2) \gtrsim -\frac{\alpha_{(t)}}{n}L(\theta_{(t)}) + \mathcal{O}\left(\alpha_{(t)}^2\right),$$

*where $\mathbf{r}(\theta) = \sum_i r_i(\theta)\phi(x_i)$.*

*Proof.* The gradient descent for the explicit parametrization is

$$\theta_{(t+1)} = \theta_{(t)} - \alpha_{(t)}\nabla_\theta L_{(t)}(\theta_{(t)}) \tag{76}$$

$$= \theta_{(t)} - \frac{\alpha_{(t)}}{k}\sum_{j:(\phi(x_j),y_j)\in\mathcal{D}_{(t)}} r_j(\theta_{(t)})\phi(x_j) \tag{77}$$

For the explicit parametrization, the decay of loss is

$$L(\theta_{(t+1)}) - L(\theta_{(t)}) \tag{78}$$

$$= \frac{1}{2}\sum_i (\langle\theta_{(t+1)},\phi(x_i)\rangle - y_i)^2 - \frac{1}{2}\sum_i (\langle\theta_{(t)},\phi(x_i)\rangle - y_i)^2 \tag{79}$$

$$= \frac{1}{2}\sum_i \langle\theta_{(t+1)} - \theta_{(t)},\phi(x_i)\rangle(\langle\theta_{(t+1)} - \theta_{(t)},\phi(x_i)\rangle + 2(\langle\theta_{(t)},\phi(x_i)\rangle - y_i)) \tag{80}$$

$$= -\frac{1}{2k}\sum_{j:(\phi(x_j),y_j)\in\mathcal{D}_{(t)}}\sum_i \langle\alpha_{(t)}r_j(\theta_{(t)})\phi(x_j) + \mathcal{O}(\alpha_{(t)}^2),\phi(x_i)\rangle(2r_i(\theta_{(t)}) + \mathcal{O}(\alpha_{(t)})) \tag{81}$$

$$= -\frac{\alpha_{(t)}}{k}\sum_{j:(\phi(x_j),y_j)\in\mathcal{D}_{(t)}}\sum_i \left(r_i(\theta_{(t)})r_j(\theta_{(t)})\langle\phi(x_i),\phi(x_j)\rangle + \mathcal{O}(\alpha_{(t)}^2)\right). \tag{82}$$

In expectation with respect to the randomness from the stochastic descent

$$\mathbb{E}[L(\theta_{(t+1)}) - L(\theta_{(t)})|\theta_{(t)}] = -\frac{\alpha_{(t)}}{n}\left\|\sum_i r_i(\theta_{(t)})\phi(x_i)\right\|_2^2 + \mathcal{O}(\alpha_{(t)}^2) \tag{83}$$

$$= -\frac{\alpha_{(t)}}{n}\left\|\mathbf{r}(\theta_{(t)})\right\|^2 + \mathcal{O}(\alpha_{(t)}^2). \tag{84}$$

We recall from our assumption that for the feature matrix $X = [\phi(x_1), \phi(x_2), \cdots, \phi(x_n)]$, $X^\top X$ has bounded norm. Thus

$$\left\|\mathbf{r}(\theta_{(t)})\right\|^2 = \left\|X(r_1(\theta_{(t)}), r_2(\theta_{(t)}), \cdots, r_n(\theta_{(t)}))^\top)\right\|^2 \leq \|X^\top X\|_2^2 \sum_{i=1}^n r_i(\theta_{(t)})^2 \lesssim L(\theta_{(t)}). \tag{85}$$

It follows that

$$\mathbb{E}[L(\theta_{(t+1)}) - L(\theta_{(t)})|\theta_{(t)}] \gtrsim -\frac{\alpha_{(t)}}{n}L(\theta_{(t)}) + \mathcal{O}(\alpha_{(t)}^2). \tag{86}$$

$\square$

## A.4 Proof of Theorem 3

**Theorem 3.** *Given a sequence of learning rate $(\alpha_{(t)})_t$, for SGD with the implicit parameterization, the expected decrease of the loss function satisfies*

$$\mathbb{E}[(L(\hat{\theta}_{t+1}) - L(\hat{\theta}_{(t)}))|\hat{\theta}_{(t)}] = -\frac{\alpha_{(t)}}{n}\left\langle P_{(t)}\mathbf{r}(\hat{\theta}_{(t)}), \mathbf{r}(\hat{\theta}_{(t)})\right\rangle + \mathcal{O}(\alpha_{(t)}^2),$$

*where $\mathbf{r}(\theta) = \sum_i r_i(\theta)\phi(x_i)$.*

*Proof.* The gradient descent for the implicit parametrization is

$$\hat{\theta}_{(t+1)} = \hat{\theta}_{(t)} - \alpha_{(t)}P_t\nabla_\theta L_{(t)}(\hat{\theta}_{(t)}) + \mathcal{O}(\alpha_{(t)}^2) \tag{87}$$

$$= \hat{\theta}_{(t)} - \frac{\alpha_{(t)}}{k}\sum_{j:(\phi(x_j),y_j)\in\mathcal{D}_{(t)}} P_t r_j(\hat{\theta}_{(t)})\phi(x_j) + \mathcal{O}(\alpha_{(t)}^2) \tag{88}$$

where

$$P_t = U_{(t)}^{-1}\left(\gamma^2 \operatorname{diag}\left(\begin{bmatrix} \|A_1^\top\hat{\theta}_{(t)}\|_2^2 \\ \vdots \\ \|A_n^\top\hat{\theta}_{(t)}\|_2^2 \end{bmatrix}\right) + I\right)U_{(t)}^{-\top} \succ 0 \tag{89}$$

$$A_k = \frac{\partial\sigma(F)_{k\dots}}{\partial F_{k\dots}} \succeq 0 \quad \forall k \in \{1, 2, \cdots, n\} \tag{90}$$

$$U_{(t)} = I - \gamma\sigma(F_{(t)}) \tag{91}$$

For the implicit parametrization, the decay of loss is

$$L(\hat{\theta}_{(t+1)}) - L(\hat{\theta}_{(t)})$$
$$= \frac{1}{2}\sum_i(\langle\hat{\theta}_{(t+1)}, \phi(x_i)\rangle - y_i)^2 - \frac{1}{2}\sum_i(\langle\hat{\theta}_{(t)}, \phi(x_i)\rangle - y_i)^2$$
$$= \frac{1}{2}\sum_i\langle\hat{\theta}_{(t+1)} - \hat{\theta}_{(t)}, \phi(x_i)\rangle(\langle\hat{\theta}_{(t+1)} - \hat{\theta}_{(t)}, \phi(x_i)\rangle + 2(\langle\hat{\theta}_{(t)}, \phi(x_i)\rangle - y_i)) \tag{92}$$
$$= -\frac{1}{2}\frac{\alpha_{(t)}}{k}\sum_{j:(\phi(x_j),y_j)\in\mathcal{D}_{(t)}}\sum_i\langle P_t r_j(\hat{\theta}_{(t)})\phi(x_j) + \mathcal{O}(\alpha_{(t)}), \phi(x_i)\rangle(2r_i(\hat{\theta}_{(t)}) + \mathcal{O}(\alpha_{(t)}))$$
$$= -\sum_i\frac{\alpha_{(t)}}{k}\sum_{j:(\phi(x_j),y_j)\in\mathcal{D}_{(t)}}\left(\langle P_t\phi(x_j), \phi(x_i)\rangle r_j(\hat{\theta}_{(t)})r_i(\hat{\theta}_{(t)}) + \mathcal{O}(\alpha_{(t)})\right)$$

We denote $\mathbf{r}(\hat{\theta}) = \sum r_i(\hat{\theta})\phi(x_i)$, then we can rewrite (92) as

$$-\sum_i\frac{\alpha_{(t)}}{k}\sum_{j:(\phi(x_j),y_j)\in\mathcal{D}_{(t)}}\langle P_t\phi(x_j), \phi(x_i)\rangle r_j(\hat{\theta}_{(t)})r_i(\hat{\theta}_{(t)}) \tag{93}$$

$$= -\frac{\alpha_{(t)}}{k} \sum_{j:(\phi(x_j), y_j) \in \mathcal{D}_{(t)}} r_j(\hat{\theta}_{(t)}) \langle P_t \phi(x_j), \mathbf{r}(\hat{\theta}_{(t)}) \rangle + \mathcal{O}(\alpha_{(t)}^2) \tag{94}$$

In expectation with respect to the randomness from the stochastic descent

$$\mathbb{E}[L(\hat{\theta}_{(t+1)}) - L(\hat{\theta}_{(t)})] = -\frac{\alpha_{(t)}}{n} \langle P_t \mathbf{r}(\hat{\theta}_{(t)}), \mathbf{r}(\hat{\theta}_{(t)}) \rangle + \mathcal{O}(\alpha_{(t)}^2). \tag{95}$$

$\square$

## A.5 Proof of Theorem 4

**Theorem 4.** *Let $r^{\|}(\hat{\theta})$ be the component of the vector of residuals parallel to the $\mathbf{1}$ direction, i.e.,*
$\mathbf{r}(\hat{\theta}) = r^{\|}(\hat{\theta})\mathbf{1} + \mathbf{r}^{\perp}(\hat{\theta})$, *then*

$$\mathbb{E}[r^{\|}(\hat{\theta}_{t+1}) - r^{\|}(\hat{\theta}_{(t)}) | \hat{\theta}_{(t)}] = -\frac{\alpha_{(t)} r^{\|}(\hat{\theta}_{(t)})}{(1 - \eta)^2} \mathbf{v}_{(t)}^{\top} \left( \eta^2 D_{(t)} + I \right) \mathbf{v}_{(t)} \frac{\|X^{\top} \mathbf{1}\|_2^2}{n} + \mathcal{O}(\alpha_{(t)}^2 + \frac{1}{1 - \eta})$$

$$\lesssim -\frac{\alpha_{(t)} r^{\|}(\hat{\theta}_{(t)})}{(1 - \eta)^2}.$$

*Proof.* Similar to the proof of Theorem 3, we can compute that

$$\mathbf{r}(\hat{\theta}_{(t+1)}) - \mathbf{r}(\hat{\theta}_{(t)}) = -\frac{\alpha_{(t)}}{k} \sum_{j:(\phi(x_j), y_j) \in \mathcal{D}_{(t)}} \sum_i \langle P_t r_j(\hat{\theta}_{(t)}) \phi(x_j), \phi(x_i) \rangle \phi(x_i) + \mathcal{O}(\alpha_{(t)}^2), \tag{96}$$

and its expectation

$$\mathbb{E}[\mathbf{r}(\hat{\theta}_{(t+1)}) - \mathbf{r}(\hat{\theta}_{(t)})] = -\frac{\alpha_{(t)}}{n} \sum_i \langle P_t \mathbf{r}(\hat{\theta}_{(t)}), \phi(x_i) \rangle \phi(x_i) + \mathcal{O}(\alpha_{(t)}^2). \tag{97}$$

Let $r^{\|}(\hat{\theta})$ be the component of the vector of residuals parallel to the $\mathbf{1}$ direction, i.e., $\mathbf{r}(\hat{\theta}) = r^{\|}(\hat{\theta})\mathbf{1} + \mathbf{r}^{\perp}(\hat{\theta})$, then

$$\mathbb{E}[r^{\|}(\hat{\theta}_{t+1}) - r^{\|}(\hat{\theta}_{(t)}) | \hat{\theta}_{(t)}] = \frac{1}{n} \mathbb{E}[\langle \mathbf{r}(\hat{\theta}_{(t+1)}) - \mathbf{r}(\hat{\theta}_{(t)}), \mathbf{1} \rangle] \tag{98}$$

$$= -\frac{\alpha_{(t)}}{n} \sum_i \langle P_t \mathbf{r}(\hat{\theta}_{(t)}), \phi(x_i) \rangle \frac{\langle \phi(x_i), \mathbf{1} \rangle}{n}. \tag{99}$$

We recall the expression of $P_t$

$$P_{(t)} = \frac{1}{(1 - \eta)^2} \mathbf{1} \mathbf{v}_{(t)}^{\top} \left( \eta^2 D_{(t)} + I \right) \mathbf{v}_{(t)} \mathbf{1}^{\top} + \mathcal{O}(\frac{1}{1 - \eta}).$$

It follows that

$$\mathbb{E}[r^{\|}(\hat{\theta}_{t+1}) - r^{\|}(\hat{\theta}_{(t)}) | \hat{\theta}_{(t)}]$$

$$= -\frac{\alpha_{(t)} r^{\|}(\hat{\theta}_{(t)})}{(1 - \eta)^2} \mathbf{v}_{(t)}^{\top} \left( \eta^2 D_{(t)} + I \right) \mathbf{v}_{(t)} \frac{\|X^{\top} \mathbf{1}\|_2^2}{n} + \mathcal{O}(\alpha_{(t)}^2 + \frac{1}{1 - \eta}) \lesssim -\frac{\alpha_{(t)} r^{\|}(\hat{\theta}_{(t)})}{(1 - \eta)^2}.$$

where for the last inequality, we used our assumption that $\|X^{\top} \mathbf{1}\|_2^2 \gtrsim n$. $\square$

## A.6 Proof of Theorem 5

**Theorem 5.** *Given $n$ data points and a batch size of $k$, the (variance)/(mean square) ratio of SGD updates is as follows.*

1. *For the explicit parametrization,*

$$\frac{\mathbb{E}[\|\Delta\theta_{(t)}\|_2^2]}{\|\mathbb{E}[\Delta\theta_{(t)}]\|_2^2} = \frac{n(k-1)}{k(n-1)} + \frac{(n-k)n}{k(n-1)} \frac{\sum_i r_i^2 \|\phi(x_i)\|_2^2}{\|\mathbf{r}(\theta_{(t)})\|_2^2}.$$

*2. For the implicit parametrization,*

$$\frac{\mathbb{E}[\|\Delta\hat{\theta}_{(t)}\|_2^2]}{\|\mathbb{E}[\Delta\hat{\theta}_{(t)}]\|_2^2} = \frac{n(k-1)}{k(n-1)} + \frac{(n-k)n}{k(n-1)} \frac{\sum_i \langle \mathbf{1}, r_i\phi(x_i)\rangle^2}{(\sum_i \langle \mathbf{1}, r_i\phi(x_i)\rangle)^2} + \mathcal{O}(1-\eta).$$

*Proof.* Let $\Delta\theta = \theta_{(t+1)} - \theta_{(t)}$. For the explicit parametrization, the change at time $t$ is

$$\Delta\theta = -\frac{\alpha_{(t)}}{k} \sum_{i:(\phi(x_i),y_i)\in\mathcal{D}_{(t)}} r_i(\theta_{(t)})\phi(x_i). \tag{100}$$

Its variance is given by

$$\mathbb{E}[\|\Delta\theta\|_2^2|\theta_{(t)}] = \frac{\alpha_{(t)}^2}{k^2}\mathbb{E}[\| \sum_{i:(\phi(x_i),y_i)\in\mathcal{D}_{(t)}} r_i(\theta_{(t)})\phi(x_i)\|^2] \tag{101}$$

$$= \frac{\alpha_{(t)}^2(n-k)}{kn(n-1)}\sum_i r_i^2\|\phi(x_i)\|_2^2 + \frac{\alpha_{(t)}^2(k-1)}{kn(n-1)}\|\sum_i r_i\phi(x_i)\|_2^2, \tag{102}$$

and its mean is given by

$$\|\mathbb{E}[\Delta\theta|\theta_{(t)}]\|_2^2 = \frac{\alpha_{(t)}^2}{n^2}\|\sum r_i\phi(x_i)\|^2. \tag{103}$$

The ratio between variance and mean is

$$\frac{\mathbb{E}[\|\Delta\theta_{(t)}\|_2^2]}{\|\mathbb{E}[\Delta\theta_{(t)}]\|_2^2} = \frac{n(k-1)}{k(n-1)} + \frac{(n-k)n}{k(n-1)}\frac{\sum_i r_i^2\|\phi(x_i)\|_2^2}{\|\mathbf{r}(\theta_{(t)})\|_2^2}.$$

Let $\Delta\hat{\theta} = \hat{\theta}_{(t+1)} - \hat{\theta}_{(t)}$. For the implicit parametrization, the change at time $t$ is

$$\Delta\hat{\theta} = -\frac{\alpha_{(t)}}{k} \sum_{j:(\phi(x_j),y_j)\in\mathcal{D}_{(t)}} P_t r_j(\hat{\theta}_{(t)})\phi(x_j), \tag{104}$$

We recall the expression of $P_t$

$$P_{(t)} = \frac{1}{(1-\eta)^2}\mathbf{1}\mathbf{v}_{(t)}^\top\left(\eta^2 D_{(t)} + I\right)\mathbf{v}_{(t)}\mathbf{1}^\top + \mathcal{O}(\frac{1}{1-\eta}).$$

For $\gamma \to 1$, we have

$$\Delta\hat{\theta} = \frac{\alpha_{(t)}}{k(1-\eta)^2}\mathbf{v}_{(t)}^\top\left(\eta^2 D_{(t)} + I\right)\mathbf{v}_{(t)} \sum_{j:(\phi(x_j),y_j)\in\mathcal{D}_{(t)}} r_j(\hat{\theta}_{(t)})\langle\mathbf{1},\phi(x_j)\rangle\mathbf{1} + \mathcal{O}\left(\frac{1}{1-\eta}\right). \tag{105}$$

Then its mean is given by

$$\mathbb{E}[\Delta\hat{\theta}] = \frac{\alpha_{(t)}}{(1-\eta)^2}\mathbf{v}_{(t)}^\top\left(\eta^2 D_{(t)} + I\right)\mathbf{v}_{(t)}\mathbf{1}\frac{\langle\mathbf{1},\sum r_i\phi(x_i)\rangle}{n} + \mathcal{O}\left(\frac{1}{1-\gamma}\right),$$

and variance is

$$\mathbb{E}[\|\Delta\hat{\theta}\|_2^2]$$

$$= \left(\frac{\alpha_{(t)}}{k(1-\eta)^2}\mathbf{v}_{(t)}^\top\left(\eta^2 D_{(t)} + I\right)\mathbf{v}_{(t)}\right)^2 \mathbb{E}\left[\left(\sum_{j:(\phi(x_j),y_j)\in\mathcal{D}_{(t)}} r_j(\hat{\theta}_{(t)})\langle\mathbf{1},\phi(x_j)\rangle\mathbf{1}\right)^2\right] + \mathcal{O}\left(\frac{1}{1-\eta}\right)$$

$$= \left(\frac{\alpha_{(t)}}{k(1-\eta)^2}\mathbf{v}_{(t)}^\top\left(\eta^2 D_{(t)} + I\right)\mathbf{v}_{(t)}\right)^2 \left(\frac{\alpha_{(t)}^2(n-k)}{kn(n-1)}\sum_i\langle\mathbf{1},r_i\phi(x_i)\rangle^2 + \frac{\alpha_{(t)}^2(k-1)}{kn(n-1)}(\sum_i\langle\mathbf{1},r_i\phi(x_i)\rangle)^2\right)$$

$$+ \mathcal{O}\left(\frac{1}{1-\eta}\right)$$

Thus the (variance)/(mean square) ratio of is given by

$$\frac{\mathbb{E}[\|\Delta\hat{\theta}_{(t)}\|_2^2]}{\|\mathbb{E}[\Delta\hat{\theta}_{(t)}]\|_2^2} = \frac{n(k-1)}{k(n-1)} + \frac{(n-k)n}{k(n-1)}\frac{\sum_i \langle \mathbf{1}, r_i\phi(x_i)\rangle^2}{(\sum_i \langle \mathbf{1}, r_i\phi(x_i)\rangle)^2} + \mathcal{O}(1-\eta).$$

$\square$

## A.7 Proof of Corollary 2 and Corollary 3

**Corollary 2.** *For the SGD with **explicit parametrization** and one-hot encoding the (variance)/(mean square) ratio of each noisy gradient update is given by*

$$\frac{\mathbb{E}[\|\Delta\theta_{(t)}\|_2^2]}{\|\mathbb{E}[\Delta\theta_{(t)}]\|_2^2} = \frac{n}{k}.$$

*Proof.* We take $\phi(x_i) = e_i$ in Theorem 5, then for the explicit parameterization

$$\frac{\mathbb{E}[\|\Delta\theta_{(t)}\|_2^2]}{\|\mathbb{E}[\Delta\theta_{(t)}]\|_2^2} = \frac{n(k-1)}{k(n-1)} + \frac{(n-k)n}{k(n-1)}\frac{\sum_i r_i^2 \|\phi(x_i)\|_2^2}{\|\mathbf{r}(\theta_{(t)})\|_2^2}$$

$$= \frac{n(k-1)}{k(n-1)} + \frac{(n-k)n}{k(n-1)}\frac{\sum_i r_i^2}{\|\mathbf{r}(\theta_{(t)})\|_2^2} = \frac{n(k-1)}{k(n-1)} + \frac{(n-k)n}{k(n-1)} = \frac{n}{k}.$$

$\square$

**Corollary 3.** *Assume the residuals are bounded such that $0 < r_{min} \le [\mathbf{r}(\hat{\theta}_{(t)})]_i \le r_{max}$ for all $1 \le i \le n$, then the (variance)/(mean square) ratio under the **implicit parameterization** is*

$$\frac{\mathbb{E}[\|\Delta\hat{\theta}_{(t)}\|_2^2]}{\|\mathbb{E}[\Delta\hat{\theta}_{(t)}]\|_2^2} \lesssim 1 + \frac{1}{k}\left(\frac{r_{max}}{r_{min}}\right)^2.$$

*Proof.* For the implicit parameterization, we have

$$\frac{\mathbb{E}[\|\Delta\theta_{(t)}\|_2^2]}{\|\mathbb{E}[\Delta\theta_{(t)}]\|_2^2} = \frac{n(k-1)}{k(n-1)} + \frac{(n-k)n}{k(n-1)}\frac{\sum_i \langle \mathbf{1}, r_i\phi(x_i)\rangle^2}{(\sum_i \langle \mathbf{1}, r_i\phi(x_i)\rangle)^2} + \mathcal{O}(1-\eta).$$

$$= \frac{n(k-1)}{k(n-1)} + \frac{(n-k)n}{k(n-1)}\frac{\sum_i r_i^2}{(\sum_i r_i)^2} \le \frac{n(k-1)}{k(n-1)} + \frac{(n-k)n}{k(n-1)}\frac{nr_{max}^2}{(nr_{min})^2} \lesssim 1 + \frac{1}{k}\left(\frac{r_{max}}{r_{min}}\right)^2.$$

$\square$

# B  Additional observation on the $O(\alpha^2)$ term and the loss surface

Theorem 1 and its proof provide the following additional observation, which is not used in the main text of this paper and is not utilized in proofs in the appendix. That is, in the proof of Theorem 1, the $O(\alpha^2)$ term is zero at any stationary points of the objective function with respect to $F$ and $w$. This can be easily seen because equation (16) and equation (34) create the $O(\alpha^2)$ terms and those $O(\alpha^2)$ terms are zero when the gradients of the loss with respect to $F$ and $w$ are zero. Thus, at any stationary points with respect to $F$ and $w$, Theorem 1 implies that $P_{(t)}\nabla L(\hat{\theta}_{(t)})$ must be zero. However, since $\lambda_{\min}(P_{(t)}) \ge c$ for some time-independent constant $c > 0$, this implies that $\nabla L(\hat{\theta}_{(t)})$ must be zero. Because $L$ is convex with respect to its argument (and every stationary point is a global minimum for a convex function), this means that every stationary point with respect to $F$ and $w$ is indeed a global minimum, without any additional assumption. Although we have not utilized this observation in any parts of our paper, this might be an useful additional observation.

Table 1: Hyper-parameters considered for each domain

| | CHAIN | FOUR ROOM | MOUNTAIN CAR |
|---|---|---|---|
| $\eta$ | $\{0.8, 0.9, 0.95, 0.975\}$ | $\{0.8, 0.9, 0.95, 0.975\}$ | $\{0.8, 0.9, 0.95, 0.975\}$ |
| $k$ | 25 | 25 | 25 |
| MAX VALUE ITERATION STEPS | 5000 | 5000 | 5000 |
| VALUE ITERATION TOLERANCE | $10^{-6}$ | $10^{-6}$ | $10^{-6}$ |
| EXPLICIT | | | |
| $\alpha$ | $\{2^i : -2 \leq i \leq 3\}$ | $\{2^i : 2 \leq i \leq 7\}$ | $\{2^i : 2 \leq i \leq 7\}$ |
| IMPLICIT | | | |
| $\alpha$ | $\{2^i : -8 \leq i \leq -3\}$ | $\{2^i : -3 \leq i \leq 2\}$ | $\{2^i : -3 \leq i \leq 2\}$ |

# C   Implementation details

## C.1   Implicit differentiation

Our experiments were implemented in JAX [4]. Back-propagation with implicit differentiation was implemented by using JAX to define the vector-Jacobian product of the constraint in combination with a the matrix-free linear solver GMRES part of the JAX package with default parameters. This allows us to solve for the gradients efficiently without needing to generate the full Jacobian of the constraint $h$. The efficient vector-Jacobian product can be thought of as a generalization of back-propagation. See the JAX documentation on autodiff for a more detailed explanation.[2]

## C.2   Hyper-parameters

See Table 1 for the hyper-parameters tried for each domain.

## C.3   Hardware

All experiments were ran on a desktop computer using a AMD Ryzen 5950X CPU and 32 GB of memory at 3800GT/s.

## C.4   Runtimes

Runtimes of individual runs are available with the code. Most runs were between 15-20 seconds for both parameterization. We didn't see much variation in runtime between domains and parameterization, with the implicit parameterization being 1-2 seconds slower in chain and four rooms, and 4 seconds slower in mountain car. It is likely that the runtime was dominated by JAX's JIT compiler and other initialization overhead. A run in this context corresponds to training with a specific combination of hyper-parameters and random seed. Each run was assigned to a single vCPU and each vCPU was never running more than 1 run at any time.

---

[2] https://jax.readthedocs.io/en/latest/notebooks/autodiff_cookbook.html

# D  Learning rate comparisons



Figure 4: Norm of the final residual on the chain MDP domain using the explicit parameterization

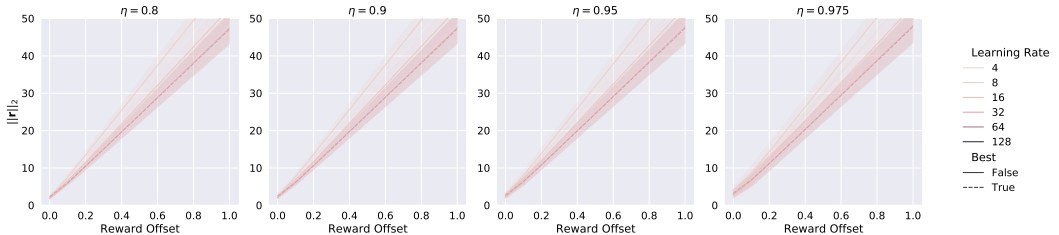

Figure 5: Norm of the final residual on the four rooms domain using the explicit parameterization

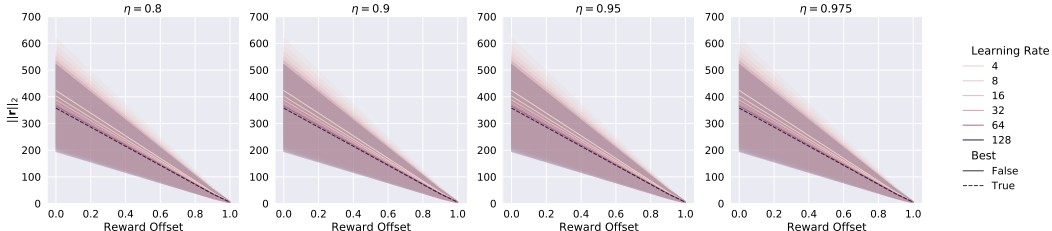

Figure 6: Norm of the final residual on the mountain car domain using the explicit parameterization

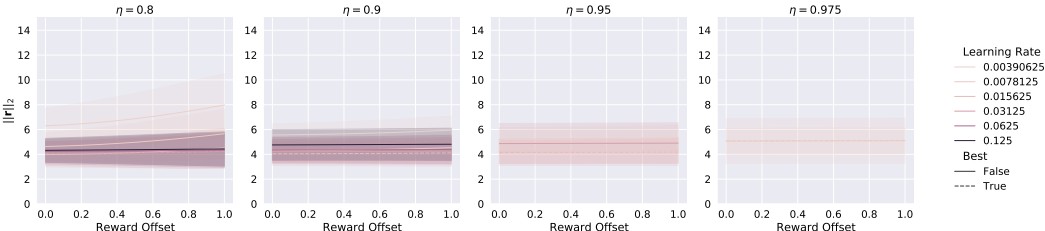

Figure 7: Norm of the final residual on the chain MDP domain using the implicit parameterization

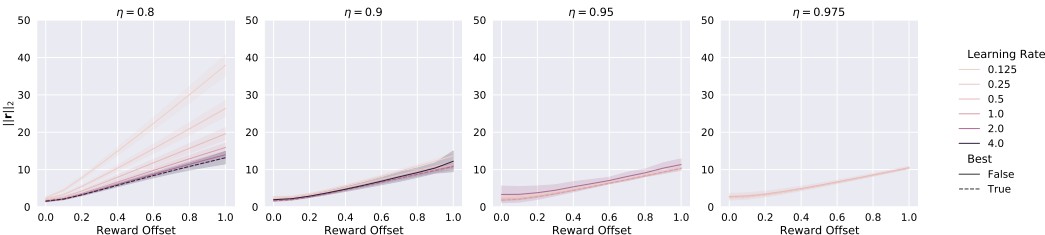

Figure 8: Norm of the final residual on the four rooms domain using the implicit parameterization

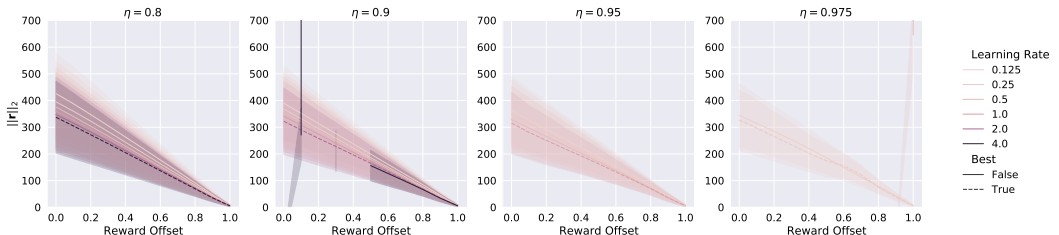

Figure 9: Norm of the final residual on the mountain car domain using the implicit parameterization

# E    Beyond the singleton action case

Although our analysis is limited to the singleton action case, we think it important to note that the gradient dynamics of the general case exhibits similar properties as the singleton case. Most notably, the $\mathbf{1}$ direction plays a similarly important role. This is due to the fact that the Jacobian $\partial h/\partial \theta$ is identical to the that of the Markov reward chain induced by the optimal policy.

We show below that the same trends seen in the singleton case are also present in the general case. The following empirical results consider the case where 4 actions are used in the internal model. Although our theoretical results don't directly apply to this case, the results here are qualitatively similar suggesting that insight from our analysis of the singleton case likely also applies to the more general case. The hyperparameters marked as "best" are the same as in the singleton case for ease of comparison.

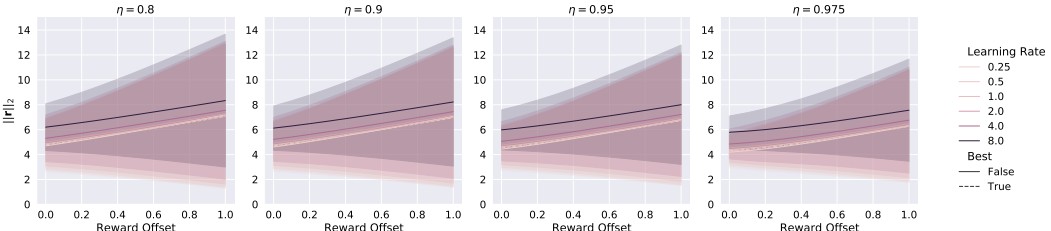

Figure 10: Norm of the final residual on the chain MDP domain using the explicit parameterization and 4 actions in the internal model

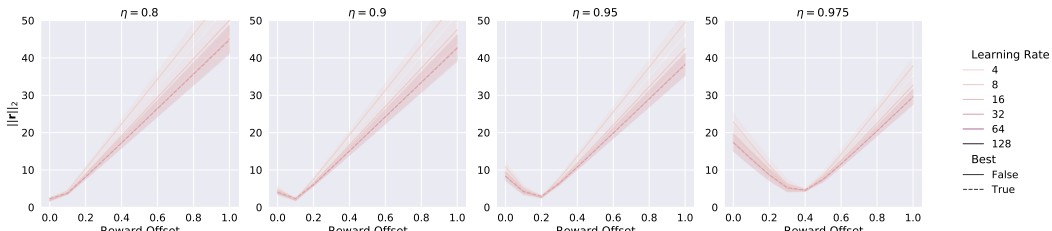

Figure 11: Norm of the final residual on the four rooms domain using the explicit parameterization and 4 actions in the internal model

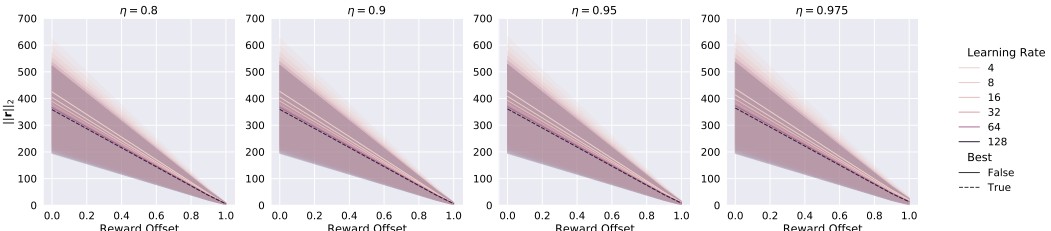

Figure 12: Norm of the final residual on the mountain car domain using the explicit parameterization and 4 actions in the internal model

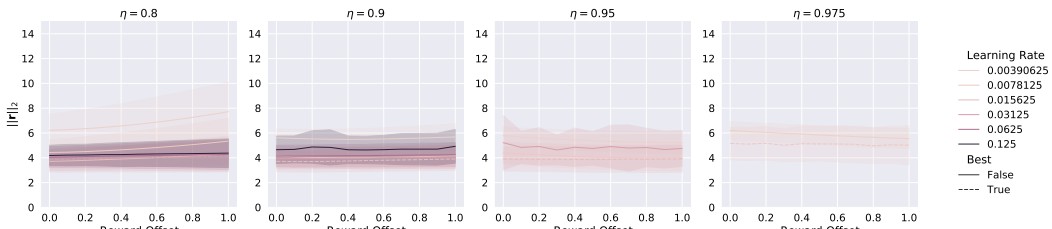

Figure 13: Norm of the final residual on the chain MDP domain using the implicit parameterization and 4 actions in the internal model

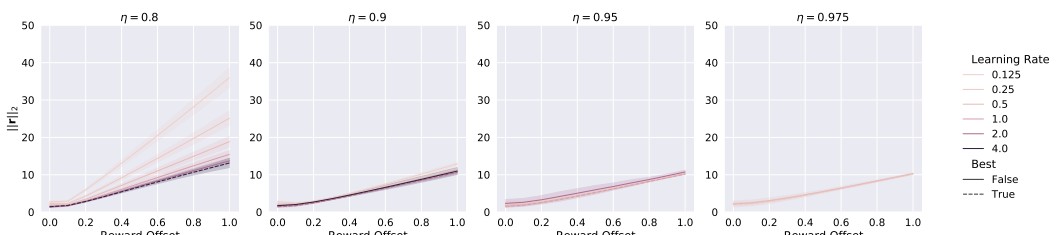

Figure 14: Norm of the final residual on the four rooms domain using the implicit parameterization and 4 actions in the internal model

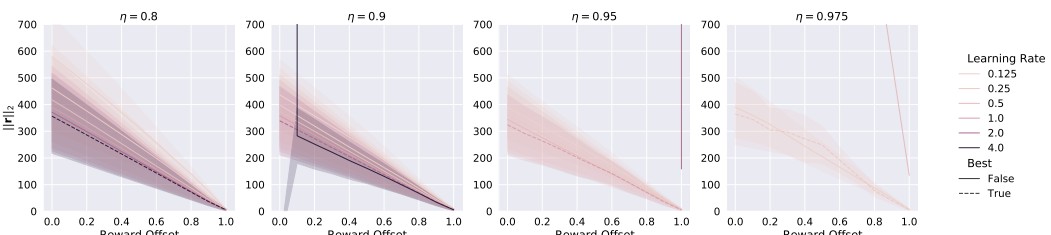

Figure 15: Norm of the final residual on the mountain car domain using the implicit parameterization and 4 actions in the internal model

## F   Empirical results for the variance-mean ratio

We include some inconclusive result plotting the terms on either side of Corollary 3. The reward offset manipulation of the domains was specifically chosen to illustrate Theorem 4 so the fact that these results doesn't allow us to conclude anything isn't surprising. Nevertheless, we include them here for the sake of transparency.

Figure 16 shows that the right-hand side of the bound does not vary much as the reward offset is applied. Consequently, we don't expect to see any strong trend in the variance-mean ratio which is

what we observe in Figure 17. However, we do see that the variance-mean ratio is constant in the explicit case, as is expected from Corollary 2.

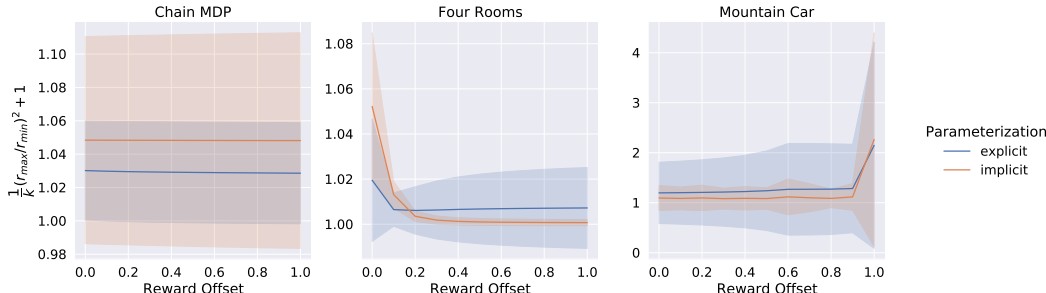

Figure 16: The right hand side of Corollary 3 at the end of training.

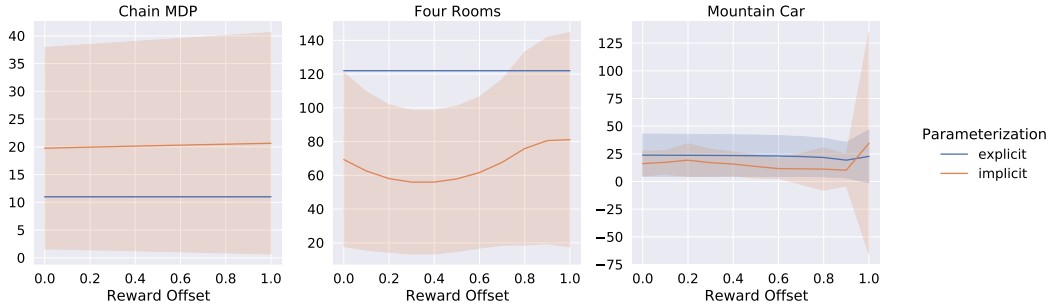

Figure 17: The variance-mean ratio of the gradient at the end of training.