# OpenReview forum: "Understanding End-to-End Model-Based Reinforcement Learning Methods as Implicit Parameterization"
_NeurIPS.cc/2021/Conference — NeurIPS 2021 Poster_

### Official Review · Reviewer_LFGm · 2021-07-13

**Rating:** 5
**Confidence:** 3

**Summary:**

The paper studies value estimation in end-to-end model-based RL methods.
The setting is simplified for analysis. They consider action-less MDP. The form of the value is assumed to be a linear combination of fixed state features \phi(s) with constant coefficients \theta. If \phi is one-hot, \theta would correspond to a vector of state values.
The implicit rep. for \theta is a log Transition matrix + reward vector and \theta is defined to satisfy the bellman condition.
The paper then studies gradient and gradient descent of regression for implicit and explicit representation.
The first main result is to show the similarity between explicit update and implicit update save for a ‘preconditioning’ matrix P (and higher order terms). They prove P positive definite and obtain global convergence from that.
In the next part they decompose the transition matrix and show how the preconditioner P behaves in the limit \eta -> 1 and its consequences (preconditioner favors 1 component).
Finally they present some experiments on toy tasks where they are showing that the 1 component of the residual is increased by a reward offset and that the final performance is less sensitive to this in the case of implicit compared to explicit.


**Limitations And Societal Impact:**

Yes

**Main Review:**

Strengths:
1. Using this analytic implicit representation for analysis is a neat idea.
2. Decomposing the transition matrix in terms of the stationary distribution is also very neat and I didn’t expect the arising form of the preconditioner.
3. The writing presents the ideas quite clearly.

Issues:
1. I believe there might be an error in corollary 3. The sum of $r_i^2$ shouldn’t cancel with the denominator which is sum of squares: $||\sum_i r_i \phi(x_i)||_2^2$ (same as in implicit case). Actually I think in this case, implicit and explicit should have the same variance/mean ratio (except for $O(1-\eta)$ term).
2. Parallel residual convergence is a main result and is the only one that is tested experimentally. However, I’m not sure why the convergence rate for the sum of residuals is important? Can’t you get 0 sum of residual by overestimating some examples and underestimating others?
3. Related to above, in Theorem 4 the authors claim faster convergence for the parallel residual by comparing the rates. This is an apples to oranges comparison though since in theorem 3 they are talking about convergence rate of loss and theorem 4 they are talking about convergence rate of sum of residual. Wouldn’t you want to compare to the rate for $\sum_i (<r_i \phi_i,1>^2$) instead?
4. Experiments: Since the theoretical results mainly regard convergence rates, it would have been good to also show learning curves wrt # of iterations.
5. The additional figures in the appendix are hard to read (why are there some learning rate runs missing for the implicit case?). There also seem to be graphical glitches on a few of them.
6. Introducing implicit differentiation seems kind of superfluous here, since the actual “implicit” rep for \theta can be easily expressed explicitly.
7. There are a few syntactic typos in the proofs (f.e. on line 95 the brackets, on line 106 duplicate $\sum_i$, line 108/109 there’s an extra 1/n out of nowhere that then disappears later)
8. This might be obvious but I think you should still state if/when the implicit rep will be as expressive as the explicit one since this affects the minima.
9. I believe you should state the $O(\alpha^2)$ term in theorem 4 as well.
10. In proof of theorem 3, near line 102, why is there no 1/n from the loss? Isn’t loss defined to have $1/|D_t|$?

Novelty/Impact:
I will first admit that optimization theory is not my area of expertise.
I think studying end-to-end model-based RL is interesting. However, I think generally the benefit that people in the broader community are more interested in when using these methods is generalization and sample efficiency rather than optimization convergence rate.
Furthermore, the simplifications used in this work are rather strong. I feel without TD learning, structural assumptions about the value targets and action selection the setting essentially reduces to supervised regression with a particular parameterization.
That being said, the paper does present some interesting results about the gradient of implicit param itself that could be built upon.

Rating:
I think the work could be an alright theoretical analysis. However, I am giving a lower rating and also lower confidence primarily because I question some of the main contributions claimed: Specifically I question the validity of corollary 3 and don’t feel like theorem 4 warrants the interpretation it is given as well as a couple technical deficiencies and sloppy writing in the proofs. But I’d be willing to raise my score if these can be clarified.

Edit:
After rebuttal, authors clarified my doubts about corollary 3 and I increased my rating to 5.
I still think the significance of the results are somewhat limited with the simplified setting.

**Time Spent Reviewing:**

7

---

> ### Author Response · Authors · 2021-08-10
> **Rebuttal**
>
> We address each individual issue below.
>
> > 1. I believe there might be an error in corollary 3
>
> For corollary 3, we specialized in the one-hot encoding case, namely $\phi(x_i)=e_i$. Then the sum $\sum_i r_i \phi(x_i)=\sum_i r_i e_i$ is given by the vector with i-th coordinate $r_i$, and its $L_2$ norm is given by $\sum_i r_i^2$, which cancels with the numerator.
>
> >2. Parallel residual convergence is a main result and is the only one that is tested experimentally. However, I’m not sure why the convergence rate for the sum of residuals is important? Can’t you get 0 sum of residual by overestimating some examples and underestimating others?
> 3. Related to above, in Theorem 4 the authors claim faster convergence for the parallel residual by comparing the rates [...]
>
> The purpose of these results is to help identify situations in which this implicit parameterization helps. It is correct that the residual can be non-zero while having no component along the [1 1 … 1]^T direction and little can be said about that situation given our results. The value of these results is in the observation that **IF** in a situation where large parts of the residual are on this component, then you should expect the implicit parameterization to converge significantly faster.
>
> > 4. Experiments: Since the theoretical results mainly regard convergence rates, it would have been good to also show learning curves wrt # of iterations.
>
> We had the learning curves in a previous version but found that they added little insight. We can add them back in the appendix if the reviewer thinks they would provide some value.
>
> > 5. The additional figures in the appendix are hard to read (why are there some learning rate runs missing for the implicit case?). There also seem to be graphical glitches on a few of them.
>
> Some of the runs have diverged (partially or fully) which causes those anomalies and the missing values.
>
> > 6. Introducing implicit differentiation seems kind of superfluous here, since the actual “implicit” rep for \theta can be easily expressed explicitly.
>
> The subject is unfamiliar to many readers and we felt it important to include to help convey the generality of this perspective. Furthermore, our experiments use implicit differentiation since our implementation supports the full MDP case. We’ve also generated some results which will be added to the appendix that considers 4 actions in order to address some of the concerns raised by other reviewers. Differentiating the 4 action case is most easily done with implicit differentiation.
>
> > 7. There are a few syntactic typos in the proofs (f.e. on line 95 the brackets, on line 106 duplicate  ∑i, line 108/109 there’s an extra 1/n out of nowhere that then disappears later)
>
> We have added the missing brackets in line 95. For line 106, the summation is over index $i$ and $j$, it follows the same argument as in (102). For line 108/109, the extra factor $1/n$ is from the definition $\mathbf{r}(\hat\theta) = r^\parallel(\hat\theta) \mathbf{1} + \mathbf{r}^\perp(\hat\theta)$. When we take the inner product of both sides with $\mathbf{1}$, the inner produce of $<1,1>$ gives a factor $n$: $<\mathbf{r}(\hat\theta), 1>=n r^\parallel(\hat\theta)$.
>
>
> > 8. This might be obvious but I think you should still state if/when the implicit rep will be as expressive as the explicit one since this affects the minima.
>
> As long as the image of the implicit function $g$ is $\mathbb{R}^n$, then they are equally expressive. This is the case for the implicit function defined by (2). We will state this explicitly in our next draft.
>
> > 9. I believe you should state the O(α2) term in theorem 4 as well.
>
> Yes, there is an extra error O(\alpha_t^2), we added it to theorem 4. But it does not affect the final expression, since in our setting the learning rate \alpha=O(1), so its square can be bounded by itself up to some multiplicative constant.
>
> > 10. In proof of theorem 3, near line 102, why is there no 1/n from the loss? Isn’t loss defined to have 1/|Dt|?
>
> For the stochastic gradient descent case we use the loss on the top of page 4, which has been normalized by $1/|D_t|$. To measure the convergence, in Theorem 3, we used change of the total loss from (3), we did not normalize it by $1/n$.

---

> > ### Comment · Reviewer_LFGm · 2021-08-15
> > **Re: Rebuttal**
> >
> > Thanks for the detailed reply. It has cleared up the points I brought up.
> > 1. This was my misunderstanding and the result is correct. I didn't take into account that we assume only 1 data point per state.
> > 2. I understand your point. However, typically for explicit rep (f.e. using linear regression) there will be a bias offset for all values and so I'm not sure how relevant this [1 1 … 1]^T direction is. Also in my original review, I meant that one can get 0 sum of residual along the 1 direction with arbitrarily bad loss (by increasingly overestimating and underestimating different states) so it's misleading to assume faster convergence in these situations just based on this.
> > 4. I think that since the main theoretical claims are about convergence rate, the comparison of the learning curves would be an important factor for providing empirical evidence.
> > 5. I see. I didn't see the diverged experiments mentioned. The presentation could still be better. It is difficult to read (at least for me).
> >
> > My major concerns with the results were addressed and I have adjusted my score to a 5.
> > I still think the significance of the paper is limited since it studies a very simplified setting and doesn't address sample efficiency/generalization which is usually the motivating factor for end-to-end model based rl as I stated in my original review.

---

### Official Review · Reviewer_CGbD · 2021-07-15

**Rating:** 5
**Confidence:** 2

**Summary:**

This paper provides a theoretical analysis of learning value functions in a Markov reward chain by SGD. They propose to analyze these methods as indirectly parameterizing weights of a linear function (with fixed features) and compare to explicitly learning the linear weights. The authors prove that, under certain assumptions, the implicit parameterization converges to the global minimum. The authors note that the gradient updates differ because the implicit parameterization preconditions the gradient. The authors then characterize the conditions under which this implicit parameterization performs better (as measured  by the variance-to-mean-square ratio) by analyzing the dynamics of the primary component of this preconditioning matrix and find that this occurs when the sign of the components all point in the same direction. The authors corroborate this theory with a series of experiments on toy environments.

**Limitations And Societal Impact:**

N/A as this paper provides mainly theoretical analysis of V-learning in Markov reward chains, which is too generic to warrant discussing societal impact

**Main Review:**

This paper studies an interesting question regarding the dynamics of V-learning in RL.

My main concern of this paper is that it is unclear how relevant this analysis is to end-to-end RL algorithms. In particular, the premise that value-based learning methods learn linear weights of fixed features requires more justification. The restriction to Markov reward chain, rather than an MDP also greatly limits the scope of the analysis.

For example, the statement, “Note that any insight in this simpler framework is likely relevant to the state-dependent model setting since the gradients differ only by the Jacobian relating model parameters’ differentials to learned parameters’ differentials, i.e., the chain rule.” seems to sweep aside the fact that this Jacobian can be arbitrarily complex for deep neural networks. Similarly, it would be useful to explain why analysis on a Markov reward chain would translate to VIN, which internally takes a max over actions.
As another example, what are the restrictions or assumptions on the constraint h and why are those assumptions realistic?

Moreover, does O(alpha^2) include terms that depend on F? If so, the abstract statement “We prove that, for a linear parametrization, gradient descent converges to global optima despite non-linearity and non-convexity introduced by the implicit representation” seems misleading as it misses an important condition. In particular, the paper proves that there is a global minimum, but I believe it requires the strong assumption that all second order terms are dominated by a quadratic. Given this strong assumption, the global convergence results are not as universal as the statement implies and are more limited in significance.

Overall, although the paper performs interesting analysis, tying this analysis back to RL algorithms and justifying the simplifying assumptions made would greatly improve the impact of this paper.

typos:
abstract: “Estimating the per-state expected cumulative rewards is a critical aspect of reinforcement learning approaches, however the experience is obtained, but standard deep neural-network function-approximation methods are often inefficient in this setting.”

**Time Spent Reviewing:**

1

---

> ### Author Response · Authors · 2021-08-10
> **Rebuttal**
>
> We address each point individually bellow.
>
> > My main concern of this paper is that it is unclear how relevant this analysis is to end-to-end RL algorithms. In particular, the premise that value-based learning methods learn linear weights of fixed features requires more justification.
>
> This fixed-feature formulation is similar to what was done in most of the value iteration network experiments. For instance, they used a one-hot encoding into a gridworld as their features to “select” the values of a specific state.
>
> > The restriction to Markov reward chain, rather than an MDP also greatly limits the scope of the analysis.
>
> This is indeed a limitation but we believe that our results provide insight on the general case.  Most of our derivations involving a single step apply to the MDP case since gradients will correspond to the gradient of the Markov reward chain (MRC) under the optimal policy (assuming no ties and an argmax policy) where the transition matrix of the induced MRC will simply be a selection of the rows of the transition for each corresponding optimal state action pair. For this reason, we expect many of the properties of the dynamics of the gradient that are present in the MRC case will also be present in the MDP case. Finally, we also ran our experiments with 4 actions as a sanity check and the results look qualitatively very similar. We will add these results in the appendix.
>
> > For example, the statement, “Note that any insight in this simpler framework is likely relevant to the state-dependent model setting since the gradients differ only by the Jacobian relating model parameters’ differentials to learned parameters’ differentials, i.e., the chain rule.” seems to sweep aside the fact that this Jacobian can be arbitrarily complex for deep neural networks.
>
> This statement was not meant to sweep aside this complexity but rather to highlight that the complexity composes. Whatever dynamics are introduced by the more complicated architecture is still altered in a similar way as the parameterization we analyse.
>
> > Moreover, does O(alpha^2) include terms that depend on F?[...]In particular, the paper proves that there is a global minimum, but I believe it requires the strong assumption that all second order terms are dominated by a quadratic.
>
> Importantly, in the proof of Theorem 1, please note that $\mathcal{O}(\alpha^2)$ is zero at any stationary points (where the gradients of the loss w.r.t. F and w are zero). This can be easily seen because eq (15) and eq (33) create the $\mathcal{O}(\alpha^2)$ terms and those $\mathcal{O}(\alpha^2)$ terms are zero when the gradients of the loss w.r.t. F and w are zero. Thus, at any stationary points, $P_{(t)}\nabla L( \hat \theta_{(t)})$ must be zero too. However, as shown in the proof of Corollary 1, $P_{(t)}$ is always positive definite for all t and its minimum eigenvalues do not approach 0. Thus, this implies that  $L( \hat \theta_{(t)})$ must be zero. Because L is convex w.r.t. its argument (and every stationary point is a global minimum for a convex function), this means that every stationary point is indeed a global minimum, without any additional assumption. In particular, we do not require that “the second order terms are dominated by a quadratic”.
>
> So, as long as the (stochastic) gradient descent converges to a stationary point, it must be a global minimum. To clarify this point, we added the above explanation in the paper.

---

> > ### Comment · Reviewer_CGbD · 2021-08-12
> > **Re: Rebuttal**
> >
> > > Importantly, in the proof of Theorem 1...In particular, we do not require that “the second order terms are dominated by a quadratic”.
> >
> > Thank you for clarifying this.
> >
> > > This statement was not meant to sweep aside this complexity but rather to highlight that the complexity composes. Whatever dynamics are introduced by the more complicated architecture is still altered in a similar way as the parameterization we analyse.
> >
> > Thank you for this clarification as well.
> >
> > I appreciate this response from the authors. This clarified some of my points of confusion. However, I think my main concerns is still unaddressed. Currently, the paper claims to provide "a theoretical account of the performance of end-to-end model-based methods," but that the relationship between this work and practical deep reinforcement learning algorithms seems too tenuous. I may be under-appreciating the theoretical contributions of this paper, but at least the writing of the paper would need to be significantly re-worked to make it clear to what extent this paper is trying to provide insight for the model-based methods discussed in the introduction. In short, I very much agree with the concerns expressed by Reviewer kLUp.

---

### Official Review · Reviewer_AqM3 · 2021-07-16

**Rating:** 6
**Confidence:** 2

**Summary:**

This paper does some theoretical study to explain the fast convergence of end-to-end model-based RL in some domains. Specifically, the authors prove global optimal convergence for the linear approximation case and study the convergence rate under certain conditions that stochastic gradient descent (SGD) with implicit representation converges substantially faster than its explicit counterpart. The authors also provide empirical results in some simple environments to back up their theory.


**Ethical Concerns:**

No ethical concerns

**Limitations And Societal Impact:**

The empirical results are conducted on toy environments and have some disparity to the actual implementation of recent model-based RL algorithms. Meanwhile, if to supplement the empirical results, the authors could do similar experiments on more complex domains, such as the Atari or Mujoco environments, and with deep RL algorithms. Though this might introduce unknown factors that affect the final convergence, any implication in such a setting might be more helpful to algorithmic works in the community.
One last concern is that the authors could have spent more paragraphs discussing related literature (both empirical and theoretical) and summarize their implications.

I don’t see any potential negative societal impact of this work.

**Main Review:**

Originality, quality, clarity of the paper:
The paper studies an observation with little formal justification in RL literature: End-to-end model-based methods are heuristically shown to be capable of better generalization in some RL domains.
There are substantial theoretical results (but I have not checked the proof carefully, since I have no prior experience in working with gradient descent dynamics). Overall, the derivation is smooth and has good logic flow, with relatively enough explanation. The definitions and quantities introduced show the thoughtfulness of the authors. Thus I think this paper is of relatively good quality.

Significance of the paper:
There are 2 main theoretical contributions. The first is showing the implicitly parameterized linear weights in the model still converges to a unique global optimal. The second is finding conditions under which the implicit parameterization exhibits significantly better properties than explicit parameterization. Besides, the empirical results seem to support the second theoretical claim. Therefore I think this paper does provide some novel insight on the principle of end-to-end model-based RL methods.

*post-rebuttal*:
 Thanks for your reply. I have read your response and my review keeps the same.

**Time Spent Reviewing:**

5

---

> ### Author Response · Authors · 2021-08-10
> **Rebuttal**
>
> We address some of the comments individually below.
>
> > Meanwhile, if to supplement the empirical results, the authors could do similar experiments on more complex domains, such as the Atari or Mujoco environments, and with deep RL algorithms.
>
> We wanted to keep the experimental results focused on illustrating our theoretical results. For this reason, we manipulate the residuals in these simple domains (through the reward offset) to show how each parameterization is affected. This allows us to make stronger conclusions than if we just looked at the relative performance of the parameterizations.
>
> > One last concern is that the authors could have spent more paragraphs discussing related literature (both empirical and theoretical) and summarize their implications.
>
> We appreciate the feedback. We were very short on space but will leverage the extra page to expand our related work and discussions.

---

### Official Review · Reviewer_g4M6 · 2021-07-16

**Rating:** 4
**Confidence:** 3

**Summary:**

This paper focuses on the area of theoretical reinforcement learning, where the authors consider the form of parameterization for the value function.  Specifically, the authors consider the case of a Markov decision process with a linear value function, and compare the explicit and implicit way of parameterizing the weights of the value function. The authors prove that using gradient descent, the implicit parameterization is equivalent to the explicit parameterization with a gradient preconditioner, which converges to the global minima. The authors then extend this result to stochastic gradient descent, and then compare the condition in which implicit parameterization gives better convergence speed. The authors provide empirical evidence in several discrete MDP environments.

**Limitations And Societal Impact:**

See main review for limitations, and N/A for social impact.

**Main Review:**

Overall I think this paper presents an interesting study of the implicit parameterization of value functions for a Markov decision process. However, I believe that there are also some limitations that need to be addressed.


Pros:

The paper is well written. The structure of the paper is well organized and the authors present the theoretical results in a step-by-step way, which makes it easy for the reader to grasp the motivation behind the theoretical results.

Being a theoretical paper, the paper also includes empirical studies in several MDP environments, making the results more intuitive.


Cons:

The choice of the specific implicit parameterization (equation 2) requires more justification. We know that setting equation 2 to zero gives us the Bellman equation for the tabular Markov reward process with rewards $w$, discount factor $\eta$ and transition probability $\sigma(F)$. The authors suggest that the internal discount factor $\eta$ is different from the true discount factor $\gamma$. Under this assumption it is clear that the ground truth transition matrix and reward vector cannot be a good fit for parameters $\sigma(F)$ and $w$. It is then unclear what the learned $\sigma(F)$ and $w$ represent. Are we looking for a different MDP dynamics and rewards with discount $\eta$ such that the value function happens to be the same as our ground truth MDP?

Moreover, the authors claim that “this implicit formulation allows us to analyze how $\psi$ and $\hat{\theta}$ relate without needing to consider the algorithm used to compute $\hat{\theta}$”. I believe this is not valid for this particular implicit parameterization, since it is well known that $\hat{\theta} = (I - \eta \sigma(F))^{-1} w$, which suggests that this implicit parameterization can be considered as just another form of explicit parameterization.



One important aspect that prevents implicit parameterization to be used in practice is the computation complexity of the implicit gradient, which the authors fail to consider in this paper. Computing the inverse Jacobian product in the implicit gradient has complexity $kd + d^2$, for each step where $d$ is the dimension of $\hat{\theta}$ and $k$ is the batch size. On the other hand normal stochastic gradient descent only has complexity $kd$. Since $d$ is usually larger than $k$, it is unclear that implicit SGD would have much advantage compared to explicit SGD given the same computation budget. Furthermore, under the one hot encoding assumption in the paper, it is not clear why SGD is even necessary if the data takes the form of empirical discounted sum of rewards, as one can obtain the exact minima of equation 3 by simply averaging all the data points of each state.


Given these concerns, it is hard to recommend the acceptance of this paper before they are addressed.



**Time Spent Reviewing:**

4

---

> ### Author Response · Authors · 2021-08-10
> **Rebuttal**
>
> We address each point individually bellow.
>
> > The authors suggest that the internal discount factor η is different from the true discount factor γ. Under this assumption it is clear that the ground truth transition matrix and reward vector cannot be a good fit for parameters σ(F) and w. It is then unclear what the learned σ(F) and w  represent.
>
> The learned parameters σ(F) and w are analogous to the transition probabilities and rewards, respectively, but are not guaranteed to have any objective meaning. Note that this isn’t a novel idea and that the value iteration networks paper have the same issue. The core idea behind end-to-end model-based methods is that the parameters relate to the ground truth only through, the value predictions or policy that they produce. Recovering the ground truth model is not an objective of end-to-end model-based methods.
>
> Furthermore, we would like to note that since this parameterization is greatly over-parameterized, there are many F and w pairs that produce  “optimal” weights $\hat{\theta}$, regardless of whether $\gamma = \eta$.
>
> > Moreover, the authors claim that “this implicit formulation allows us to analyze how ψ and θ^ relate without needing to consider the algorithm used to compute θ^”. I believe this is not valid for this particular implicit parameterization [...]
>
> We contrast taking an implicit perspective with other approaches to end-to-end model-based methods, e.g., value iteration networks, which differentiate through the computation graph of the solver. We prefer analysing an analytical form independent of the many design decisions that define the solver. Examining derivatives through implicit differentiation allows us to do this, independent of the implementation of the solver itself. Could the reviewer elaborate on why the existence of an explicit formulation invalidates our statement?
>
> > One important aspect that prevents implicit parameterization to be used in practice is the computation complexity of the implicit gradient, which the authors fail to consider in this paper.
>
> First, we would like to emphasize that this work doesn’t propose a new method to do end-to-end model-based RL. The formulation proposed here is to help build insight about how these methods behave and how they can be thought of as a particular choice of parameterization. This allows us to analyze how this choice alters the gradient dynamics. The computational complexity is not relevant in our theoretical analysis of the gradient dynamics.
>
> Second, there are many examples of implicit differentiation being used in recent work (see [1] for an overview) so the computational issues may not be as dire as the reviewer expects them to be. Solving for this inverse can be done only using matrix-vector products of the jacobian which can be efficiently implemented with automatic differentiation. The cost of each matrix-vector product is then simply the cost of backpropagating through the constraint which, in general, is much cheaper than materializing the full quadratic sized jacobian. Furthermore, this Jacobian is often only defined on a subset of the parameters, further improving the complexity.
>
> [1] http://implicit-layers-tutorial.org/
>
> > Furthermore, under the one hot encoding assumption in the paper, it is not clear why SGD is even necessary if the data takes the form of empirical discounted sum of rewards, as one can obtain the exact minima of equation 3 by simply averaging all the data points of each state.
>
> The purpose of our empirical results are to illustrate our theory. Since our theory analyzes the dynamics of SGD, our experiments focus on SGD. We do not claim to improve on state of the art in any of these simple domains.

---

> > ### Comment · Reviewer_g4M6 · 2021-08-16
> > **Re: Rebuttal**
> >
> > Thanks for the detailed response!
> >
> > >The core idea behind end-to-end model-based methods is that the parameters relate to the ground truth only through, the value predictions or policy that they produce. Recovering the ground truth model is not an objective of end-to-end model-based methods.
> >
> > If the proposed method does not intent to recover the original transition probability and rewards, then it is not clear to me why studying this particular parameterization is important, since there are many other (over-parameterized) parameterizations that can also be used to represent the value function.
> >
> > > Could the reviewer elaborate on why the existence of an explicit formulation invalidates our statement?
> >
> > Since we have an analytical form of the explicit formulation, the exact gradients computed through the explicit formulation and the implicit formulation are identical regardless of what solvers are used. If inexact solvers are used to compute the inverse in either the implicit or explicit form, then the errors are introduced by the solvers themselves, which are independent from the specific choice of formulation and are also not analyzed in this paper. Therefore, I believe that the use of implicit differentiation is not necessary in paper.
> >
> > > First, we would like to emphasize that this work doesn’t propose a new method to do end-to-end model-based RL. The formulation proposed here is to help build insight about how these methods behave and how they can be thought of as a particular choice of parameterization. This allows us to analyze how this choice alters the gradient dynamics. The computational complexity is not relevant in our theoretical analysis of the gradient dynamics.
> >
> > If the authors believe that this paper does not try to propose a new method to do end-to-end model-based RL but instead focuses on a specific choice of parameterization, then it would be necessary to justify why this choice of parameterization is important. I believe that merely showing that SGD converges faster under this parameterization is not sufficient, as there are many other parameterizations or optimization methods that can achieve faster convergence rates than SGD.
> >
> > Overall, I will keep my score of the paper after reading the rebuttals and other reviews.

---

### Official Review · Reviewer_kLUp · 2021-07-29

**Rating:** 3
**Confidence:** 4

**Summary:**

This paper presents a theoretical analysis of two different ways of parameterising a linear value function approximator. They consider an 'explicit' parameterisation, in which the weights connecting the state features to value estimate are parameterised directly, and an implicit parameterisation, which attempts to use knowledge of the properties of a Markov process to inform a more structured parameterisation.

The authors analyse the learning dynamics of these two parameterisations under stochastic gradient descent, and draw some conclusions about the properties of RL environments that might favour the implicit parameterisation. Finally, they carry out some empirical investigations on offline datasets of trajectories from several different environments.


**Limitations And Societal Impact:**

There are no anticipated negative societal impacts from this work.

**Main Review:**

The authors attempt to motivate their work as a theoretical investigation into model-based methods that model transition probabilities and rewards. In particular, they attempt to draw a close parallel to "value iteration networks", which is a particular form of model-based method.

Despite the undoubted importance of investigations into model-based planning, I am not convinced that this is really what this work achieves. In particular, their method is really a different parameterisation of a purely model-free method - as far as I can see there is no internal policy improvement or value iteration step based on a model, which is what a model-based method would be aiming to achieve. For example, value iteration networks use repeated applications of a convolutional network with max pooling as a form of differentiable "end-to-end" value iteration in order to inform an improved policy.

Additionally, their implicit parameterisation seems really most applicable to a discrete-state Markov process (relying on a vector of rewards and an explicit transition matrix as components of its parameterisation), which is not typically the case in more complex environments.

Their theoretical analysis provides some interesting insights into how environments with reward offsets can impact learning in this setting. However, I am unconvinced that this would be an issue for modern algorithms that use temporal differences and baselines, in part precisely to help improve learning in such contexts. In the setting with no reward offset, it seems that both parameterisations perform similarly well in their empirical results.

In summary, despite an admirable intention to investigate an important area, I am not convinced this paper makes sufficient contributions to the community in its current form.


**Time Spent Reviewing:**

3

---

> ### Author Response · Authors · 2021-08-10
> **Rebuttal**
>
> We address each point individually bellow.
>
> > In particular, their method is really a different parameterisation of a purely model-free method - as far as I can see there is no internal policy improvement or value iteration step based on a model, which is what a model-based method would be aiming to achieve. For example, value iteration networks use repeated applications of a convolutional network with max pooling as a form of differentiable "end-to-end" value iteration in order to inform an improved policy.
>
> Both our formulation and value iteration networks should be thought of as “model-free” methods. Neither method explicitly fits a model to observed data. Instead, they fit predictions based on some internal model. For instance, the core empirical results found in the value iteration network paper uses a supervised learning approach for fitting a policy derived from the predicted q-values. In contrast, we focus on a regression setting and fit the values directly.
>
> We identified properties induced only by the form of these predictions. These predictions take on a very similar form even in the presence of planning since the optimal value function is just the value function of the Markov reward chain induced by the optimal policy. We believe that our insights are applicable to the non-singleton action case. As a sanity check, we re-ran our experiments with 4 internal actions and observed the same qualitative results. We will include these results in the revised appendix.
>
> > Additionally, their implicit parameterisation seems really most applicable to a discrete-state Markov process (relying on a vector of rewards and an explicit transition matrix as components of its parameterisation), which is not typically the case in more complex environments.
>
> We don’t propose this specific formulation be used as an algorithm to solve complex problems. This formulation is meant as a theoretically tractable proxy to help build our understanding of how taking gradients through models alters the dynamics of gradient descent.
>
> > In the setting with no reward offset, it seems that both parameterisations perform similarly well in their empirical results.
>
> The objective of this work is not to motivate end-to-end model-based methods but to provide some insight on how such a parameterization affects the gradient dynamics. The focus of our experiments was to show how the rates of convergence change when we manipulate the residuals (through the reward offset). We believe our empirical results illustrate our theory well. The purpose was never to argue implicit > explicit.

---

> > ### Comment · Reviewer_kLUp · 2021-08-19
> > **Response to authors**
> >
> > I would like to thank the authors for their response to my comments and for the thoughtful discussion with the other reviewers.
> >
> > Their response does not change my original assessment of the paper, which is that the maths and results shown are sound, but that I do not believe that its conclusions are strong enough to warrant recommending acceptance. The results around the gradient dynamics of the implicit versus explicit parameterisations are likely of interest to a specialist audience, but do not (in my judgement) lead to a significantly better understanding of or improvement on existing "end-to-end model-based" (i.e. motivated parameterisations of model-free) methods.

---

### Official Review · Reviewer_EkYa · 2021-08-02

**Rating:** 3
**Confidence:** 4

**Summary:**

This paper proposes a theoretical framework to analyze model-based end-to-end reinforcement learning. In particular, the authors focus on the MBRL with a linear approximation of the value function, and within the Markov reward chain setting (with singleton action). Theoretical results show that MBRL in an end-to-end way is actually doing implicit parameterization and convergence analysis.


**Limitations And Societal Impact:**

This paper only has results based on a simplified model, Markov reward chains (MRC), which is different from the MDP in the RL literature in the sense that, MRC doesn't involve the action selection. In the meanwhile, the obscureness of gradient updates in this paper makes the readers hard to understand the significance of the work, neither using nor applying the results in other researches. In particular, how to sample the datapoints is not discussed in the paper, therefore I'm convinced by the results can explain the dynamic nature of RL.

**Main Review:**

### Originality

The main contribution of this paper is to provide a theoretical analysis of MBRL. With the implicit differentiation theorem, the authors link the implicit parameterization of the learned model and the linear feature weights, which looks interesting to me. The most interesting result of this paper, in my opinion, is that introducing reward and transition models is an implicit way of preconditioning in the gradient flow.

### Quality

As claimed by the authors, all main results in this paper are derived under the Markov reward chain framework. However, what I can see in the main part, e.g., Theorem 1 and Corollary 1/2, they're all analyzed under finite-sum square-error and no direct link to the common reinforcement learning scenario. For example, how the value function is estimated and updated? How does the agent update/improve the policy? What's the end-to-end MBRL algorithm the agent is performing? Based on this, I cannot agree with the authors' conclusion that

> ..., we believe our results are likely to generalize to the full MDP setting since ...

I would like to explain this point with a few more details, which lead to my concerns regarding the quality of this paper.

1. The constraint Eq. 2 appears quite confusing to me. I'd like to see more justification on why this constraint can hold. It seems to me that it can be derived from the Bellman equation, but it leads to another problem: the optimality of $\hat{\theta}$ is not identical to that of $\psi$ when $g()$ is non-linear. For example, when the image of $g()$ is a subset of $\hat{\theta}$, the optimal $\psi$ may not be obtained. For this reason, I'm not convinced by the proof of Corollary 1 (and 2).

2. Another issue on implicit differentiation is about the assumption on $\frac{\partial g}{\partial \psi}$: it should exist, but may not hold in many cases, e.g., when linear piece-wise activations like ReLU are involved in a neural network.

3. The analysis of finite-sum square error (Eq. 3) is not clear in MBRL. Does this equation apply at each timestep $t$, or at the end of each episode? For either case, how does the agent collect the datapoints (from the real environment or the model)? ... I think such ambiguity comes from the clarity issue of presentation. See the *Clarity* section for more details.
    * I'm trying to understand this in two cases: a) when the gradient updates in Eq. (4) are performed once per timestep; b) when the steps in Eq. (4) are performed until converging in each step.
    * In case a), the loss function $L$ and the datapoints should change along the time, which vanishes the assumption of the proof (a finite-sum result).
    * In case b), we can still conclude that the value prediction converges to a global minimum at the step, but it relies on the fact that such value prediction can apply on all timesteps, which is not discussed in the paper.
    * Note that this also holds for the Markov reward chain discussed in the paper: the datapoints collected at step $t$ could be different from that in other steps.

4. I'm also confused about the explicit parameterization scheme in the paper. One reason might be from the above point of clarity issue, and the other one is: it seems to me that explicit parameterization is not using any models, so it's actually a model-free RL (MFRL)? Considering the fact that MBRL takes more calculations (planning steps) during training, it's not surprising to me that the implicit parameterization converges faster than explicit parameterization.

### Clarity

This paper has some issues with writing and organization. For example:

1. This paper doesn't have a clear formal definition of the Markov reward chain that can be usually found in the RL literature. As a result, readers would find it hard to understand why such a model can help for MBRL analysis.
2. The authors don't provide a clear context on the gradient steps, especially when the steps are performed. As discussed in the *Quality* section, this would result in many confusions and weakens the conclusions of this paper.
3. Given the results from the finite-sum minimization, the authors should provide clarifications on how this relates to MBRL.

### Significance

The authors provide a convergence analysis to understand the learning of MBRL. However, as discussed above, the results are mostly derived from finite-sum with unknown sampling methods, it's hard for readers and other researchers to use the results for the MBRL study, either in theory or in practice.

**Time Spent Reviewing:**

20

---

> ### Author Response · Authors · 2021-08-10
> **Rebuttal**
>
> We address each point individually bellow.
>
> > The constraint Eq. 2 appears quite confusing to me. I'd like to see more justification on why this constraint can hold. It seems to me that it can be derived from the Bellman equation, but it leads to another problem: the optimality of θ^ is not identical to that of ψ when g() is non-linear. For example, when the image of g() is a subset of θ^, the optimal ψ may not be obtained. For this reason, I'm not convinced by the proof of Corollary 1 (and 2).
>
> It is not quite clear to us what the reviewer is trying to say here. Since the loss depends on $\psi$ only through $\hat{\theta}$, then a $\psi$ which induces an optimal $\hat{\theta}$ is optimal, by definition. Our results show that a globally optimal $\hat{\theta}$ is achieved asymptotically and therefore an optimal $\psi$. It is possible that the reviewer’s concern originates from the fact that the mapping isn’t one-to-one, e.g., there are several psi that generate the same $\hat{\theta}$, but this is inconsequential in this context since we focus on understanding the properties of the linear function learned.
>
> The reviewer also expresses the concern that the image of $g$ defined by (2) doesn’t cover $\mathbb{R}^n$. The proof that the image does in fact cover $\mathbb{R}^n$ is straightforward. For any $\theta \in \mathbb{R}^n$, pick any $F$ such that $(I - \eta \sigma(F))$ is full rank then find $w$ by solving $\theta = (I - \eta \sigma(F))^{-1} w$. Such a $w$ exists since the matrix $(I - \eta \sigma(F))$ is full rank.
>
> > Another issue on implicit differentiation is about the assumption on ∂g∂ψ: it should exist, but may not hold in many cases, e.g., when linear piece-wise activations like ReLU are involved in a neural network.
>
> The function $g$ must be continuous in a neighborhood around its input, $\psi$. This is true regardless of whether the function is explicitly or implicitly defined. For a function implicitly defined using a constraint, as in our formulation, this is the case under mild assumptions, such as the constraints being continuous in the same neighborhood and the Jacobian being invertible at that point. Assumptions of continuity like these are used even with “normal” differentiation of ReLU. Furthermore, our results do not consider ReLU so these technicalities aren’t relevant in our analysis.
>
> > The analysis of finite-sum square error (Eq. 3) is not clear in MBRL. [...] so it's actually a model-free RL (MFRL)? [...] Given the results from the finite-sum minimization, the authors should provide clarifications on how this relates to MBRL.
>
> We do not consider “conventional" model-based RL in which the model is estimated separately using a prediction-based loss. We focus on a class of methods usually referred to as “end-to-end model-based RL” (see the value iteration networks paper for a more detailed description of this type of approach). The core idea of these methods is to formulate a model like one would in a model-based method, but instead these methods will fit the output of solving for, say, the optimal value function of this model directly (as opposed to fitting transitions). It is correct that this can be thought of as a “model-free” approach but, as we highlight in this work, these methods can have special properties not found in “conventional” estimators with model-free methods.
>
> > In case a), the loss function  and the datapoints should change along the time, which vanishes the assumption of the proof (a finite-sum result).
>
> Our results consider a supervised learning approach to policy evaluation. The data used in the loss is fixed.
>
> > This paper doesn't have a clear formal definition of the Markov reward chain that can be usually found in the RL literature. As a result, readers would find it hard to understand why such a model can help for MBRL analysis.
>
> As stated in line 81, a Markov reward chain is equivalent to an MDP with a single action.
>
> > The authors don't provide a clear context on the gradient steps, especially when the steps are performed. As discussed in the Quality section, this would result in many confusions and weakens the conclusions of this paper.
>
> Could the reviewer clarify this question? We consider gradient descent (GD) and stochastic gradient descent (SGD) on a loss function. Updates are done sequentially and we use subscripts to identify parameters at different iterations (see paragraph after (3)).

---

> > ### Comment · Reviewer_EkYa · 2021-08-19
> > **Response to authors**
> >
> > I would like to thank the authors for their responses to my concerns and answering my questions for clarification.
> >
> > I still have concerns on the ambiguity of "end-to-end model-based reinforcement learning" in this paper. Though a model is usually trained separately, it's not uncommon to combine planning and learning in modern MBRL researches, and the authors do cite these previous that I appreciate it. However, any MBRL algorithm including the "end-to-end" ones, should include policy evaluation, policy improvement and model updates components, which can be done together. Unfortunately, the full scheme is not clear in this paper but just a few gradient update steps, and the relation to the MBRL is unclear.
> >
> > Moreover, I believe the result of Markov reward chain in the paper is not straightforward to extend to the MDPs (in fact, a more typical way in the literature is to have results in k-arm bandits then extend them to MDPs). I might be missing some existing work using Markov reward chains as a simplified MDP, but I'm not convinced that results for Markov reward chain are important to understand "end-to-end" MBRL. One particular reason is, Markov reward chain misses two essential components in MBRL: policy evaluation and policy improvement.
> >
> > For clarity issue, I think the authors should significantly improve the writing to make the readers clear. It takes me quite a lot of efforts to find the assumptions of the introduced functions. I appreciate the authors efforts to clarify some of my concerns. However, I'm still puzzled by the comparison of explicit parameterization vs. implicit parameterization, and I'm not sure why such proposed method is an MBRL, not "model-free". I agree with Reviewer kLUp on this point.
> >
> > Overall, I think the paper at the current form is not ready for contributions to the community, and I will keep my original score and assessment of this paper.

---

### Author Response · Authors · 2021-08-10
**Thank you for the feedback**

We would like to first thank the reviewers for their thoughtful feedback, comments, and corrections. We've opted to answer each individually but wanted to highlight the following addition to the final version of the paper. We will add to the appendix similar figures to the existing ones showing results for the case when using a 4 action MDP (instead of a Markov reward chain). The results look qualitatively similar and suggest that our results might generalize to the MDP case.

---

### Decision · Program_Chairs · 2021-09-28

**Decision:**

Accept (Poster)

**Comment:**

The paper aims at presenting a theoretical investigation of the gradient dynamics and the convergence properties of model-based RL methods. In order to make the analysis theoretically tractable, the authors analyze these methods by looking at a simplified setting: training a linear value function approximator using two formulations, an implicit one, i.e. parameterizing weights of a linear function with fixed features, and explicitly learning the linear weights.

The first contribution of the paper is providing proof that under some assumptions the implicit parameterization converges to the global minimum, and showing that the gradient updates differ in the two settings because the implicit parameterization induces a preconditioning on the gradients. A second contribution is in further analysing the properties of the preconditioning. The theoretical findings are finally verified experimentally on small scale problems using offline datasets, rendering the theoretical contributions easier to understand.

Whilst the paper attempts to address a very interesting and relevant problem in the field, and the submission indeed presents a stimulating study, all six reviewers have struggled to find a clear and strong connection between the insights proposed by the authors and "a theoretical account of the performance of end-to-end model-based methods" and lamented lack of substantial (practical) results. Therefore, I fear the paper in the current format will have limited impact at NeruIPS, and draw little follow up from this audience.

I hope that the authors will not give up on this work, and will consider either submitting to a venue with a more specialist audience, or acting on the specific recommendations some of the reviewers have provided, e.g. reframing the work to make it easier for the readers to draw connections between its results and the model-based methods discussed in the introduction.

**Consistency Experiment:**

NeurIPS has a long history of experimentation. In 2014, NeurIPS ran an experiment in which 10% of submissions were reviewed by two independent committees to quantify the randomness in the review process. This year, we repeated a variant of this experiment to see how the quality of the review process has changed over time.  This paper was part of the experiment and was therefore assigned to two committees (consisting of reviewers, an Area Chair, and a Senior Area Chair) that reached independent decisions.  If both committees made the same recommendation, this recommendation was followed. If a single committee recommended acceptance, the paper was accepted (with the exception of a few cases in which the other committee identified what we considered a fatal flaw, e.g., an error in a key result).

This copy’s committee reached the following decision: **Reject**

The other committee assigned to the paper recommended **Accept (Poster)**.  You can find the other set of reviews, along with any follow up discussion with the authors here:
https://openreview.net/forum?id=xj2sE--Q90e